# Decoding the influence of anticipatory states on visual perception in the presence of temporal distractors

Freek van Ede [1], Sammi R. Chekroud [1], Mark G. Stokes [1,2] & Anna C. Nobre [1,2]

Anticipatory states help prioritise relevant perceptual targets over competing distractor stimuli and amplify early brain responses to these targets. Here we combine electroencephalography recordings in humans with multivariate stimulus decoding to address whether anticipation also increases the amount of target identity information contained in these responses, and to ask how targets are prioritised over distractors when these compete in time. We show that anticipatory cues not only boost visual target representations, but also delay the interference on these target representations caused by temporally adjacent distractor stimuli—possibly marking a protective window reserved for high-fidelity target processing. Enhanced target decoding and distractor resistance are further predicted by the attenuation of posterior 8–14 Hz alpha oscillations. These findings thus reveal multiple mechanisms by which anticipatory states help prioritise targets from temporally competing distractors, and they highlight the potential of non-invasive multivariate electrophysiology to track cognitive influences on perception in temporally crowded contexts.

---

[1] Oxford Centre for Human Brain Activity, Wellcome Centre for Integrative Neuroimaging, Department of Psychiatry, Warneford Hospital, University of Oxford, Oxford OX3 7JX, UK. [2] Department of Experimental Psychology, University of Oxford, Anna Watts Building, Radcliffe Observatory Quarter, Woodstock Road, Oxford OX2 6GG, UK. These authors contributed equally: Mark G. Stokes, Anna C. Nobre. Correspondence and requests for materials should be addressed to F.v.E. (email: frederik.vanede@ohba.ox.ac.uk)

I n a world in which the amount of information that reaches our senses is increasing by the day, it is becoming increasingly relevant to understand the mechanisms by which our brains extract and prioritise information that is most relevant to current goals. Foreknowledge of what, where or when relevant events are likely to occur enables the instantiation of anticipatory neural states that provide key determinants of such prioritisation[1–3], and it has long been recognised that such anticipatory states amplify early brain responses to perceptual targets. In fact, such effects provided the first clear evidence in humans that modulatory effects of anticipatory attention occur early during sensory processing[4–6]. Yet, despite a long tradition, vast literature and sustained interest in this line of research[7–9], it has remained unclear whether anticipation actually amplifies the amount of information defining the identity of the perceptual target in these early sensory brain responses. Building on recent progress on multivariate decoding of visual orientation information from high temporal resolution magneto-encephalography and electroencephalography (M/EEG) measurements[10–15], we tackled this issue directly and reveal that anticipatory states also amplify stimulus-identity information contained in early visual EEG responses.

Multivariate decoding with high temporal resolution additionally enabled us to individuate neural information linked to target vs. competing distractor items occurring within the temporal window of attentional competition. While the neural mechanisms that prioritise inputs that compete in space have received ample scientific investigation[16–19], the mechanisms by which the human brain accomplishes such prioritisation for inputs that compete in time remains far less well understood. This is in part because conventional human neuroimaging approaches have been hampered either by insufficient temporal resolution (as with functional Magnetic Resonance Imaging; fMRI), or by the presence of additive responses when stimuli occur in fast temporal succession (as with classical event-related-potential (ERP) analyses). By combining stimulus orientation decoding analyses with high temporal resolution EEG measurements, we reveal that anticipatory states not only enhance neuronal target representations, but also delay the interference on these target representations caused by temporally adjacent distractors, thereby possibly providing a protected temporal window for extended target analysis.

## Results

**Anticipation facilitates perception in face of distraction.** Thirty healthy human volunteers performed a visual orientation reproduction task in which the presence/absence of preparatory auditory cues and temporally adjacent visual distractors were orthogonally manipulated (Fig. 1a; Methods for details). Auditory cues, when present, indicated that a target would follow after 500 ms. Because our main research questions regard largely unexplored territory, we deliberately used such simple (but highly effective) temporal warning cues. While we will refer to the influence of these cues as anticipation, we acknowledge up front that this type of anticipation likely involves a mix of involuntary increases in arousal and voluntary orienting of attention in time[20].

The left panel of Fig. 1b depicts average reproduction errors and highlights the utility of the cue in reducing distractor interference. While we found no cueing benefit on performance in distractor-absent trials ($t_{(29)} = 0.3021$, $p = 0.7647$, $d = 0.056$), reliable cueing benefits occurred in distractor-present trials (i.e., lower reproduction errors to cued vs. uncued targets), which interacted with inter-stimulus-interval (ISI; $F_{(2,58)} = 13.784$, $p = 1.266e^{-5}$, $\eta_p^2 = 0.322$). Planned comparisons confirmed a moderate cueing benefit at the 20-ms ISI ($t_{(29)} = -3.674$,

$p = 0.001$, $d = -0.671$), a large benefit at the 100-ms ISI ($t_{(29)} = -6.488$, $p = 4.214e^{-7}$, $d = -1.184$), but no longer any benefit when distractors followed targets at an ISI of 200 ms ($t_{(29)} = -0.011$, $p = 0.991$, $d = -0.002$). Cueing benefits were also significantly larger in distractor-present compared to distractor-absent trials, both at 20-ms ISI ($t_{(29)} = -3.779$, $p = 7.267e^{-4}$, $d = -0.67$) and at 100-ms ISI ($t_{(29)} = -7.314$, $p = 4.683e^{-8}$, $d = -1.335$). (Note that ISIs of 20, 100 and 200 correspond to SOAs of 70, 150 and 250 ms, respectively). Although reports could only be programmed and executed after the target-probe interval, we also observed robust cueing benefits on reaction times (RT; Fig. 1b, right panel). However, these appeared much more generic. Planned comparisons now also revealed a cueing benefit in distractor-absent trials ($t_{(29)} = -6.869$, $p = 1.515e^{-7}$, $d = -1.254$), and similar effects on distractors-present trials, at least at the 100-ms ISI ($t_{(29)} = -6.633$, $p = 2.852e^{-7}$, $d = -1.211$) and the 200-ms ISI conditions ($t_{(29)} = -3.899$, $p = 5.253e^{-4}$, $d = -0.712$). No significant cueing benefit was observed at the 20-ms ISI condition ($t_{(29)} = -1.635$, $p = 0.113$, $d = -0.299$), although the interaction with ISI was not significant either ($F_{(2,58)} = 2.833$, $p = 0.067$, $\eta_p^2 = 0.089$).

Because we had anticipated (based on prior piloting) that the 100-ms ISI would be particularly effective, we had deliberately used this ISI in the vast majority (80%) of distractor-present trials. Unless otherwise specified, all reported analyses below were performed exclusively on this set to ensure sufficient trial numbers per condition.

To further interrogate the behavioural performance data, we ran a mixture-modelling analysis[21] quantifying the precision of the orientation reproduction reports, alongside the proportion of reports classified as a target report, a distractor report ('swapping error'), or a random guess. Figure 1c shows these parameters as a function of cue and distractor presence. For precision, we observed a significant main effect of cue presence, with higher precision for cued compared to uncued trials ($F_{(1,29)} = 7.514$, $p = 0.01$, $\eta_p^2 = 0.206$), as well as a significant main effect of distractor presence, with lower precision for distractor-present trials ($F_{(1,29)} = 68.569$, $p = 3.959e^{-9}$, $\eta_p^2 = 0.703$). Although the interaction between cue presence and distractor presence was not significant ($F_{(1,29)} = 1.767$, $p = 0.194$, $\eta_p^2 = 0.057$), planned comparisons revealed that cues increased precision in distractor-present ($t_{(29)} = 3.598$, $p = 0.001$, $d = 0.657$), but not in distractor-absent trials ($t_{(29)} = -0.877$, $p = 0.388$, $d = 0.160$). In addition, we observed that cues increased the number of target reports in the distractor-present ($t_{(29)} = 4.688$, $p = 6.035e^{-5}$, $d = 0.856$), but not the distractor-absent trials ($t_{(29)} = -1.054$, $p = 0.301$, $d = -0.192$), this time also marked by a significant interaction between cue and distractor presence ($F_{(1,29)} = 27.619$, $p = 1.247e^{-5}$, $\eta_p^2 = 0.488$). Complementing this increase in target reports in distractor-present trials, we also found that cues reduced the number of distractor reports in these trials ($t_{(29)} = -4.360$, $p = 1.492e^{-4}$, $d = -0.796$). Finally, we observed a significant interaction between cue and distractor presence also for the proportion of guess reports ($F_{(1,29)} = 9.006$, $p = 0.006$, $\eta_p^2 = 0.237$), where cues significantly reduced the proportion of guesses in distractor-present ($t_{(29)} = -2.336$, $p = 0.027$, $d = -426$), but not in distractor-absent trials ($t_{(29)} = 1.054$, $p = 0.301$, $d = 0.192$).

The impact of the cue on performance in our task is further visualised in Fig. 1d, showing response distributions aligned to the target and distractor orientations. When there are no distractors (left panel), response distributions look very similar between cued and uncued trials. However, in face of temporal distractors (right panel), cues increase the proportion of target responses (solid lines), while reducing the proportion of distractor responses (dashed lines). Collectively, these data thus

demonstrate that cues can facilitate perception by overcoming temporal distractors.

**Decoding to individuate target and distractor processing.** Our main aim was to investigate the influence of the preparatory cues and temporally competing distractors on the amount of sensory information contained in the EEG responses regarding the identity (orientation) of target and distractor stimuli. To this end, we applied a time-resolved decoding approach. Before turning to the influence of the preparatory cues, we explain our decoding approach and highlight its utility in individuating and tracking in time both target and distractor representations.

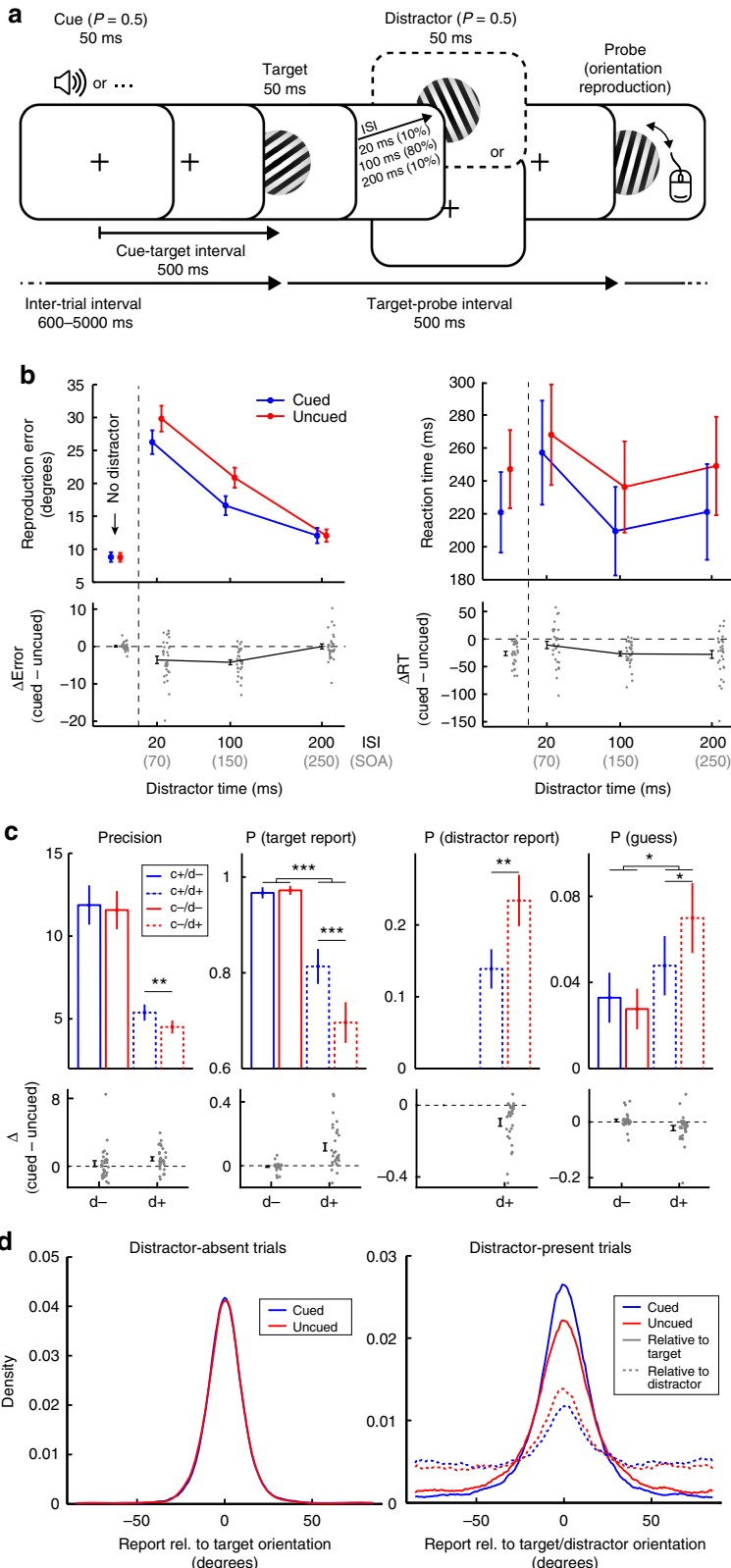

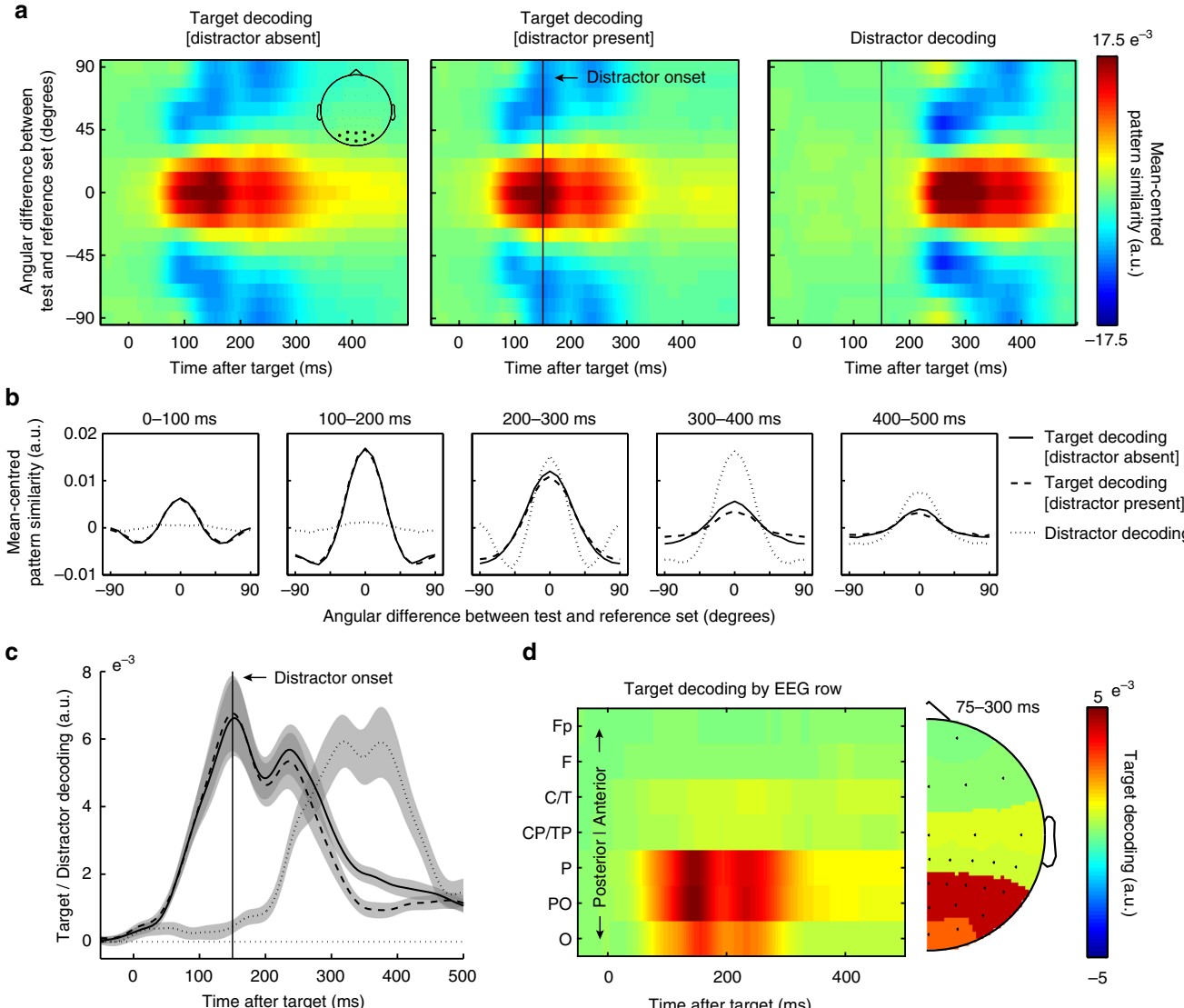

**Fig. 2** Time-resolved EEG orienting decoding of targets and distractors. **a** Time-resolved orientation tuning profiles. Data represent the mean-centred pattern similarity (quantified using the Mahalanobis distance) between the test trials and the reference trials, as a function of the angular difference between test and reference trials (y axis). The inset in the leftmost panel highlights the eight electrodes that were used for the orientation decoding analysis. **b** Average tuning profiles for the data in panel **a**, in five successive time windows. **c** Time courses of the corresponding summary decoding statistic (Methods for details). **d** Time-resolved decoding (summary statistic) as a function the EEG electrode row used for decoding. Topography plot to the right shows the same data in a more conventional manner whereby the value in each electrode indicates how well the row to which that electrode belongs is able to decode target orientation. Error bars represent ±1 s.e.m calculated across participants ($n = 30$)

Per time point, we calculated the multivariate Mahalanobis distance (using electrodes as dimensions) between the left-out trial (the test trial) and all other trials (the reference or training trials)— in which the target or distractor orientation was at a particular angular difference from the test trial. By evaluating this multivariate distance metric for a range of angular differences between test and reference trials, we were able to reconstruct orientation tuning profiles (as in ref. [22]). We mean-centred (across all angular differences) the obtained distances and inversed the data such that lower distances (reflecting higher pattern similarity) are plotted as positive values (as in ref. [22]).

**Fig. 1** Task design and behavioural performance. **a** Visual orientation reproduction task with preparatory auditory cues and visual distractors. Participants reproduced the orientation of the visual target grating using a computer mouse. In half the trials, targets were preceded by an auditory warning cue. Targets could be followed by no distractors, or by a visual distractor at one of three ISIs (20, 100, 200 ms). Target-probe intervals and inter-trial intervals were drawn independently of cue and distractor presence. **b** Average orientation reproduction errors (in degrees) and reaction times (in ms) for cued and uncued trials, as well as their difference, as a function of distractor presence and ISI. **c** Mixture-modelling parameters as a function of cue and distractor presence. 'c+' for cue present, 'd−' for distractor absent, and so on. **d** Response distributions centred on the target and the distractor orientation. To ensure sufficient trial numbers, we only considered distractor-present trials in the 100-ms ISI condition in panels **c** and **d**. Error bars represent ±1 s.e.m. calculated across participants ($n = 30$). ISI, inter-stimulus-interval; SOA, stimulus-onset-asynchrony. *$p < 0.05$; **$p < 0.01$; ***$p < 0.001$. Scattered data points in panels **b** and **c** show individual participant data. To increase visibility, individual data points were slightly jittered in the horizontal plane

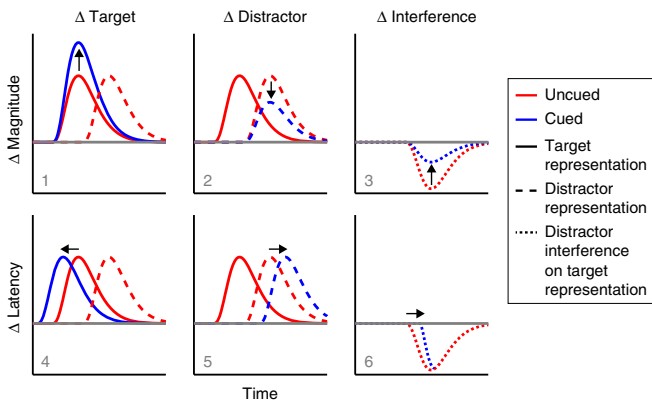

**Fig. 3** Schematic of ways in which anticipation may prioritise relevant over irrelevant sensory inputs that compete in time. We consider six non-mutually exclusive scenarios. Anticipation can influence the target representation (left), the distractor representation (middle) or the interference on the target representation when distractors are present compared to absent (right). This influence can be manifest either as a change in magnitude ('representation strength') or a change in latency ('representation timing'). We find evidence for scenarios 1 (increased target identity decoding) and 6 (delayed distractor interference on the target identity decoding)

Figure 2a shows the time-resolved tuning profiles separately for all targets without a distractor (left), all targets with a distractor (middle), as well as for all distractors (right). Clearly, neural responses evoked by the targets (left two panels) and the distractors (right panel) are more similar to other targets/distractors that have similar orientations (red), relative to other targets/distractors that have dissimilar orientations (blue). Thus, despite that fact that the latter two conditions always contained two stimuli in close temporal succession, both target and distractor identities (which were drawn independently of each other across trials), were decodable with high temporal resolution from the EEG.

Figure 2b shows the tuning profiles associated with the data in panel a, averaged over successive 100 ms time windows. To capture this tuning profile (orientation decoding) in single metric per time point, we simply multiplied the (mean-centred) tuning profiles with an inverted cosine function and averaged the result along all angular differences between test and reference trials (as in ref. [22,23]). Figure 2c shows the time courses of this summary statistic associated with the tuning profiles in Fig. 2a. We report on this summary statistic in all further analyses.

To concentrate our decoding analysis on visual activity, we limited the decoding analysis to data from the eight most posterior electrodes (inset Fig. 2a), which also showed the largest ERP. To further substantiate the visual origin of the orientation decoding, we additionally ran this analysis separately for each of the electrode rows. As shown in Fig. 2d (for all targets), this confirmed a predominantly posterior (putatively visual) origin.

**Anticipation boosts target coding and distractor resistance.** We next evaluated EEG orientation decoding as a function of cue and distractor presence. We considered six (non-mutually exclusive) scenarios by which anticipatory states may help prioritise relevant over irrelevant sensory inputs that compete in time (Fig. 3). As we detail below, we found evidence in support of scenarios 1 (enhanced target decoding) and 6 (delayed distractor interference).

Figure 4a depicts time-resolved orientation decoding for each of the experimental conditions, for both targets and distractors.

Cluster-based permutation statistics[24] were used to evaluate the main effects of cueing (cued vs. uncued trials), distractor presence (distractor-present vs. absent trials) and their interaction (i.e., the cueing effect in distractor-present vs. absent trials), while circumventing the multiple-comparisons encountered along the time axes. Although we below state the time ranges of the significant clusters as they were observed in the observed (non-permuted) data, we note that this cluster-based permutation test does not warrant inferences regarding significant time ranges, as it evaluates whether the compared conditions are 'exchangeable' or not—and, for this evaluation, it considers the full time range[24,25].

First, we observed a main effect of cue presence (light blue), as reflected in better orientation decoding for cued compared to uncued targets (cluster $p = 0.006$, cluster-interval in non-permuted data: 118 to 248 ms post target). This is in line with scenario 1 in Fig. 3. In contrast, we found no significant cueing effect on distractor decoding (if anything, we observed a numerical increase, rather than a decrease, arguing against scenario 2 in Fig. 3). Second, we observed a main effect of distractor presence (pink), as reflected in reduced target decoding for distractor-present compared to distractor-absent trials (i.e., distractor interference; cluster $p = 0.004$, cluster-interval in non-permuted data: 262 to 414 ms post target). Finally, we observed an interaction between cue and distractor presence (green; cluster $p = 0.03$, cluster-interval in non-permuted data: 196 to 268 ms post target). This effect was constituted by a larger cueing benefit for distractor-present trials, or, equivalently, a larger distractor interference for cue-absent trials. While we note that the main effect of distractor presence on target decoding was maximal in the time window in which distractor decoding itself was also maximal, the interaction effect on target decoding was maximal in the time window in which the distractor decoding emerged (Fig. 4a). All three effects were again largely confined to the posterior electrode rows (Fig. 4c), thus enhancing their 'plausibility'[26]. We also note that these effects were largely invariant to our choice of data smoothing (Supplementary Fig. 1) or baselining (Supplementary Fig. 2).

Figure 4b shows the time courses of distractor interference on target decoding (i.e., target decoding in distractor-present minus distractor-absent trials), and suggests that the observed interaction may be best understood as a delayed distractor interference effect (scenario 6 in Fig. 3). While both cued and uncued trials ultimately reach a similar level of distractor interference (unlike scenario 3 in Fig. 3), the onset of this interference appears delayed in cued trials. To further quantify this delay, we estimated the latencies at which the cued and uncued interference effects first reached the value associated with 50% of the maximal interference value (averaged over both conditions), and used a Jackknife approach (as described in ref. [27]) to evaluate this delay statistically. This confirmed a $77 \pm 18.57$ ms (mean ± SE) delay in cued compared to uncued trials (Jackknife $t_{(29)} = -4.145$, $p = 6.74e^{-5}$). Moreover, although we initially selected 50% of the maximum interference value for this analysis, it is reassuring to note that similar statistics were obtained when estimating latencies from values ranging anywhere from 10 to 70% of the maximum interference value (right panel Fig. 4b).

These results suggest that cues are particularly useful for protecting target analysis from distraction. To test this possibility more directly, we performed another complementary analysis. We reasoned that, if cues protect target analysis from interference, then target representations in distractor-present trials with a cue should more closely resemble distractor-absent trials than should distractor-present trials without a cue (i.e., in distractor-present trials, cues should make the target representation appear more as if there was no distractor). To test this

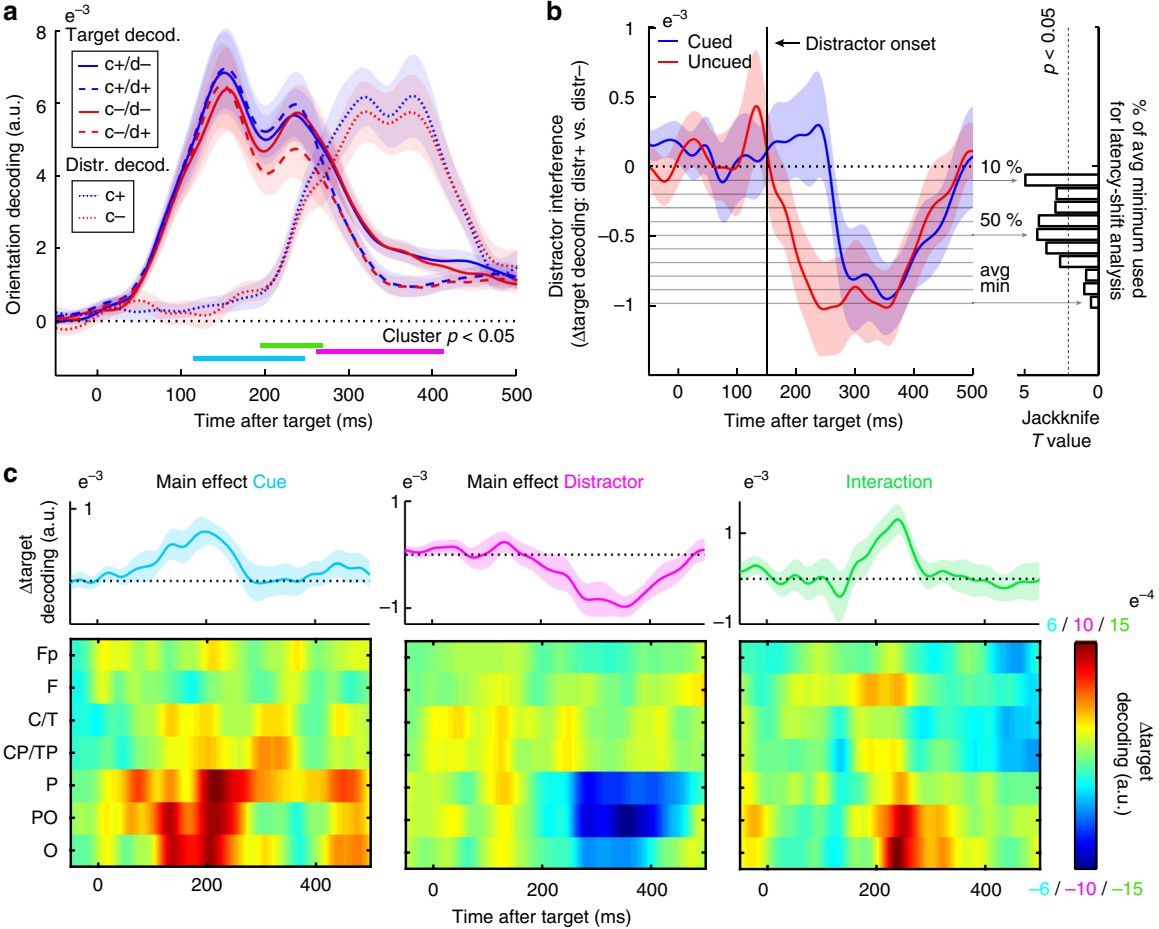

**Fig. 4** Anticipation increases target identity information and delays distractor interference in early visual EEG responses. **a** Time courses of target and distractor orientation decoding (summary statistic) as a function of cue presence (blue for cued, 'c+'; red for uncued, 'c−') and distractor presence (solid for distractor-absent, 'd−'; dashed for distractor-present, 'd+'). Horizontal lines mark where the clusters of the contrasts that survived cluster-based permutation statistics were observed in the non-permutated data for the main effects of cue presence (light blue), distractor presence (pink), as well as their interaction (green). All clusters involve target decoding; no significant cueing effect cluster was observed for distractor decoding. **b** Time courses of the distractor interference effect on target decoding. Distractor interference is quantified as the difference in target decoding for distractor-present vs. absent trials. Right panel shows Jackknife T values for latency differences between cued and uncued trials at thresholds ranging from 10 to 100% of the maximal interference effect. Maximal interference was calculated as the lowest value in the average of the cued and the uncued trials (denoted 'avg min'). **c** Main and interaction effects as a function of time and electrode row. The interaction is expressed as the difference between cue-present vs. absent trials in distractor-present vs. absent trials. Upper plots show decoding based on the same channels as in **a** (see Fig. 2a). Shadings represent ±1 s.e.m. calculated across participants ($n = 30$)

prediction, we re-evaluated target orientation decoding, but this time only included distractor-absent trials (irrespective of cue condition) in our reference ('training') set. This confirmed that target orientation could be better decoded from cued compared to uncued distractor-present trials (Supplementary Fig. 3).

Although limited trial numbers for the 20-ms and 200-ms ISI distractor conditions hampered statistical sensitivity for quantifying cueing benefits on decoding, we did observe qualitatively similar patterns in these conditions whereby, descriptively, distractor interference immediately after the respective distractor time appeared attenuated by the cue (Supplementary Fig. 4). This appeared particularly clear in the 20-ms ISI condition for which we also observed a similar cueing benefit on behavioural accuracy. In this condition we further noted substantially larger distractor interferences in target decoding (Supplementary Fig. 4) in further agreement with the behavioural performance data.

In contrast to scenarios 4 and 5 in Fig. 3, Fig. 4a showed no evidence for a cueing effect on the latencies of either target or distractor decoding alone (target: Jackknife $t_{(29)} = 0.251$,

$p = 0.299$; distractor: Jackknife $t_{(29)} = 0.342$, $p = 0.316$). We additionally ran a cross-temporal decoding analysis whereby we only included uncued trials in our reference sets and tested decoding performance on cued trials. Decoding was always best when reference and test times corresponded (Supplementary Fig. 5), thus providing further evidence that the EEG 'orientation code' does not appear to shift forward (for targets) or backward (for distractors) in time with cueing.

Because our decoding was based on the broadband visual responses, a natural question is whether the observed cueing effects on target decoding and distractor resistance may simply be carried over from amplified ERP responses in cued trials (for example, by virtue of higher signal-to-noise ratio). When evaluating ERP amplitudes (Fig. 5a, c), we did also observe a main cueing effect that occurred at a similar time window as the main cueing effect on target decoding (putatively reflecting amplification of the classic N1 potential). Interestingly, however, across our 30 participants, the magnitude of this cueing effect on the ERP appeared uncorrelated with the magnitude of the cueing

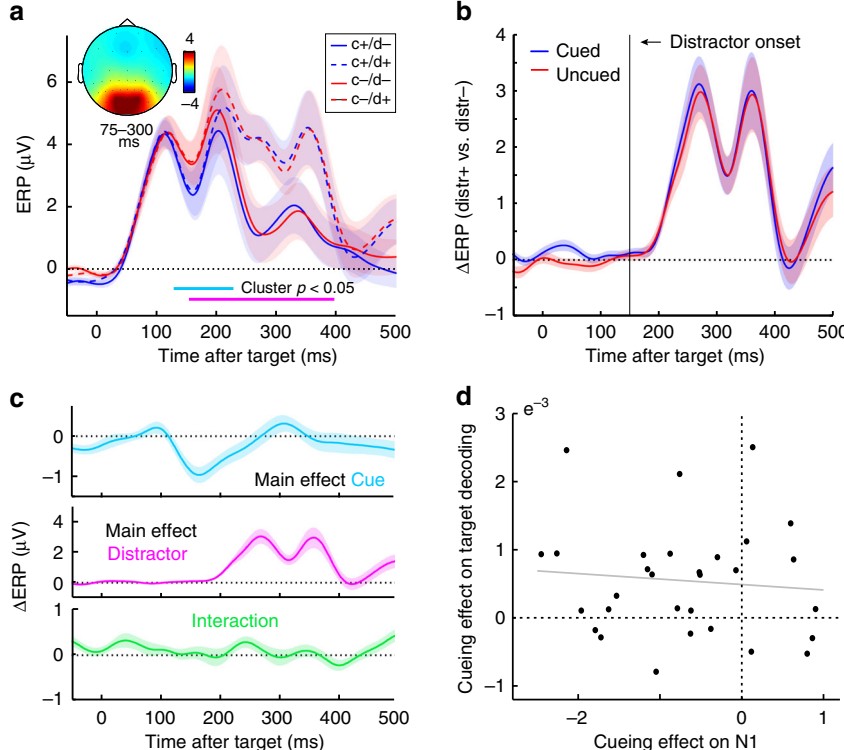

**Fig. 5** Event-related potentials as a function of cue and distractor presence. **a** Time courses of the ERP in the selected eight most posterior channels as a function of cue and distractor presence (cf. Fig. 4a). Horizontal lines mark where the clusters of the contrasts that survived cluster-based permutation statistics were observed in the non-permutated data for the main effects of cue presence (light blue) and distractor presence (pink). No significant interaction effect was observed. Data are baseline corrected by a 250-pre-target baseline. Topography inset shows the visually evoked ERP component between 75–300 ms post-target (collapsed across all trial types). **b** Difference ERP for distractor-present vs. absent trials, as a function of cue presence. **c** Time courses of the main effects of cue and distractor presence, as well as their interaction. The interaction is expressed as the difference between cue present vs. absent trials in distractor-present vs. absent trials (cf. Fig. 4c). **d** Scatter plot showing absence of a correlation between the main cueing effect on the ERP (130–229 ms post-target), and the main cueing effect on target orientation decoding (118 to 248 ms post-target). Individual data points represent individual participants ($n = 30$)

effect on decoding (Fig. 5d; $r = -0.093$, $p = 0.624$). In further contrast to the decoding results, we also did not observe an interaction between cue and distractor presence on the ERP that could account for the increased distractor resistance observed in decoding (Fig. 5b, c) or behaviour (Fig. 1). We did observe a clear effect of distractor presence (Fig. 5a-c), which was expected given summation of target and distractor evoked responses (this contrasts starkly with the decoding analysis where we could individuate target from distractor representations).

**Attenuated alpha oscillations facilitate target prioritisation.** A key marker of attentional orienting in human M/EEG measurements is the anticipatory attenuation of 8–14 Hz alpha oscillations in relevant sensory brain areas[28–33]. Here we link such brain states in posterior electrodes to increased target decoding (across participants), as well as distractor resistance (across trials), thereby corroborating (using orthogonal analyses) the above described influences of the anticipatory cues.

Figure 6a shows the time-resolved and frequency-resolved difference in spectral power between cued and uncued trials, averaged over all posterior electrodes. Following a transient increase in low-frequency power with a frontal-central topography (left topography Fig. 6a) that likely reflects processing of the auditory cue, we observed a decrease in 8–14 Hz power with a more posterior topography (right topography Fig. 6a). This likely reflects the instantiation of an 'attentional brain state'. This state

appears to emerge before target onset (in line with above references), although in our data it becomes most prominent during target and distractor processing (likely as consequence of our relatively short cue-target interval). Corroborating this attentional interpretation, we found that those participants who showed the largest cue-induced alpha attenuation, also showed the largest cue-induced reductions in RT (expressed as the RT-ratio between cued and uncued trials), yielding a positive correlation (Fig. 6c; cluster $p = 0.005$, cluster-interval in the non-permuted data: $-180$ to $220$ ms post target; frequency range: 6 to 11 Hz). This correlation has a clear posterior topography (topography Fig. 6b).

To address whether the cue-induced modulation of this brain state is also related to the cue-induced amplification of target decoding, we also correlated each time-frequency sample in Fig. 6a with the participant-specific magnitude of the main cueing effect on target decoding. Figure 6c shows the resulting correlation map. Participants with a stronger cue-induced alpha attenuation also have a larger cueing effect on decoding; resulting in a negative correlation (cluster $p = 0.024$, cluster-interval in the non-permuted data: $-20$ to $300$ ms post target; frequency range: 6 to 11 Hz). This correlation also has a clear posterior topography (topography Fig. 6c), and appears not to be driven by outliers (scatter plot Fig. 6c).

We additionally evaluated the relation between alpha states and target decoding across trials. We focused on all uncued trials (where spontaneous variability is expected to be largest, and where there is no contamination with cue processing) and sorted

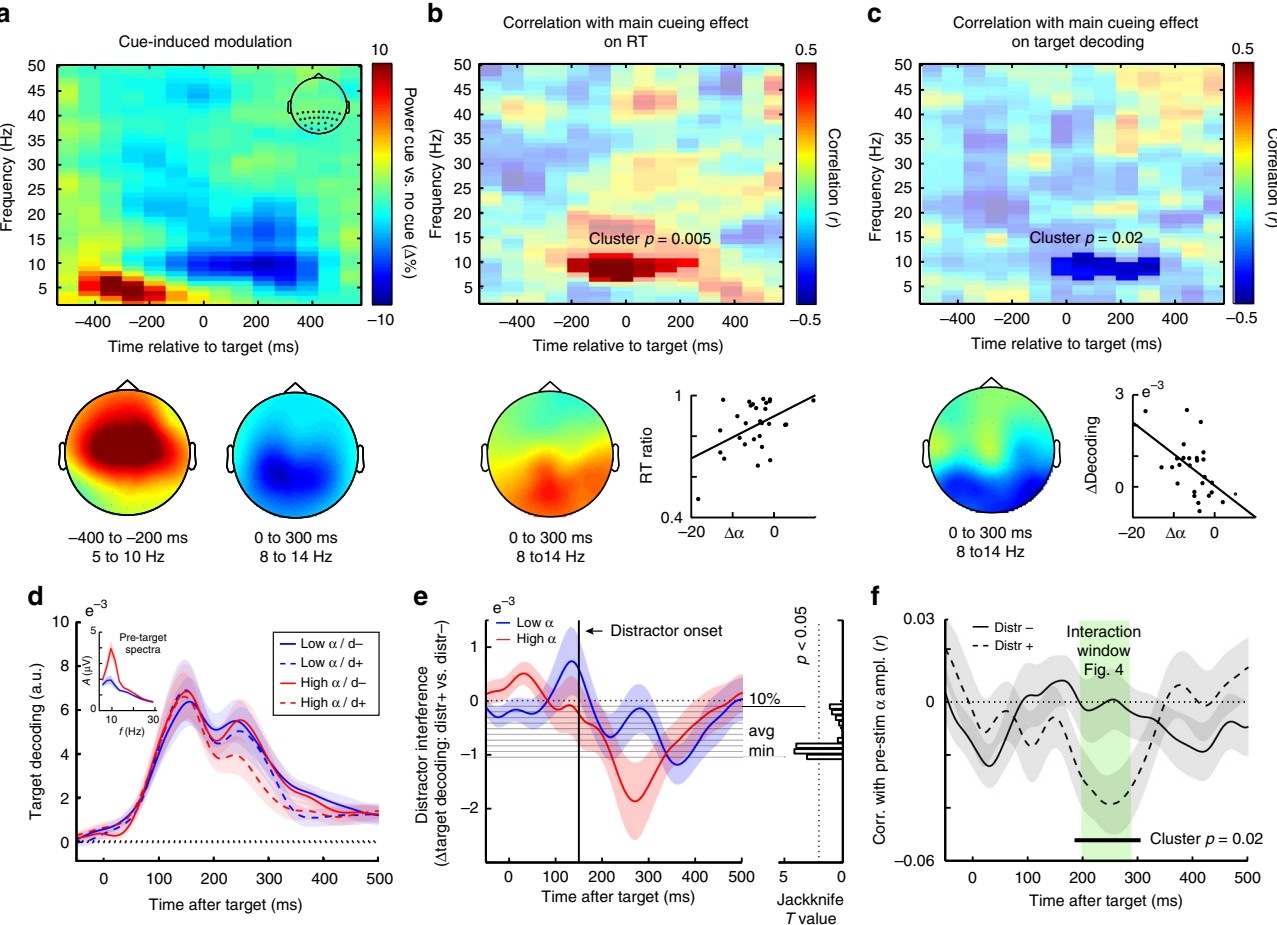

**Fig. 6** Attenuated posterior alpha oscillations predict enhanced target decoding (across participants) and distractor resistance (across trials). **a** Time-frequency plot of the cue-induced modulation in spectral power, expressed as a percentage change (i.e., ((cued−uncued)/(uncued))×100). Data from all posterior electrodes marked in the inset in the right top. Topographies show modulations from 5 to 10 Hz in the interval between −400 to −200 ms (left) and from 8 to 14 Hz in the interval between 0 and 300 ms post-target (right). Topographies were scaled according to the same colorbar as the time-frequency plot. **b** Time-frequency plot of the correlation (across participants) of the cue-induced modulation with the magnitude of the main cueing effect on reaction time (RT; expressed as a ratio between cued and uncued RTs). **c** Similar to panel **b**, except showing the correlation with the main cueing effect on target decoding (averaged over 118 to 248 ms post-target; see Fig. 4a) The participant-specific magnitudes of the alpha modulation used for the scatter plots in panels **b** and **c** were extracted from the significant time-frequency clusters and only serve to show the underlying distributions. **d** Time courses of target decoding in uncued trials as a function of distractor presence and pre-target alpha amplitude (median split). Trials were sorted by alpha amplitude averaged over all posterior channels in the 500 ms pre-target interval. Inset shows associated pre-target spectra. **e** Distractor interference time courses as a function of pre-target alpha state. Same conventions as for Fig. 4b. **f** Time courses of the trial wise correlation between pre-target alpha amplitude and target decoding, separately for distractor-present and absent trials. Shadings represent ± 1 s.e.m. calculated across participants (n = 30). The green shaded band in panel **f** highlights the similarity of the alpha-dependent decoding effect with the cue-dependent decoding effect (the interaction effect) in Fig. 4

the trials by alpha amplitude in the 500-ms pre-target window. Figure 6d shows target decoding as a function of both pre-target alpha amplitude (median split; inset for associated spectra) and distractor presence, while Fig. 6e shows the corresponding distractor interference time courses (cf. Fig. 4a, b) as a function of pre-target alpha state. This separation by pre-target alpha state suggests that not only cues (as in Fig. 4), but also spontaneous states of low-amplitude alpha oscillations can reduce the immediate interference by the distractor. Figure 6f quantifies this relation between pre-target amplitude and target decoding on the basis of a trial-wise correlation. This confirmed that the influence of pre-target alpha amplitude appears particularly prominent when distractors were present, whereby attenuated alpha states (lower amplitudes) are associated with a 'protective' effect (better target decoding) (Fig. 6f, cluster p = 0.02, cluster-interval in the non-permuted data: 186–307 ms post-target). This is highly reminiscent of the interaction

effect observed between cue and distractor presence (Fig. 4a). In fact, we noted a strikingly similar time window between both effects (as highlighted in Fig. 6f). This analysis thus further marks the utility of anticipatory states—either after cues, or as reflected in 'spontaneously' attenuated alpha oscillations—on preserving target decoding in the face of temporal distractors.

These correlations cannot be trivially explained by an increase in signal variance due to higher alpha amplitude. To take away this potential concern, all presented decoding analysis were performed on the time-domain signal from which we had removed the 8–14 Hz band using a band-stop filter (we confirmed that qualitatively similar results were obtained when no filter was applied). At the same time, several recent studies have shown that not only target location[34] but also orientation[35] can be decoded also from topographical maps of alpha amplitude. While we could confirm such alpha-based orientation decoding in our data, this appeared less robust and did not yield clear conditional

differences as a function of cue and distractor presence (Supplementary Fig. 6).

We also investigated potential cue-induced resetting of oscillatory phase[36], as well as potential relations between pre-target alpha phase and target decoding, as the phase of alpha oscillations may also critically shape perception[37]. However, beyond a clear phase-reset in the lower frequencies (which, again, most likely reflected the auditory ERP), we did not find compelling evidence for anticipatory phase-alignment following the cue, nor did we observe compelling associations between pre-target phase and target decoding.

**Correspondence between decoding and behavioural performance.** In the simplest perceptual task one would expect the quality of target decoding to correlate with behavioural performance on a trial-by-trial basis. In our task, however, many factors are likely to influence behavioural performance, of which the perceptual processing that takes place during initial encoding is but one. Behavioural performance was likely influenced by many additional factors that are not well captured by the early EEG responses that we focused on (such as post-target lapses in short-term memory, post-target changes of mind as to which item was the target, interference by the probe, motoric errors, etc.). This may explain why we were not able to demonstrate compelling and consistent correlations between the trial-by-trial variability in the magnitude of target decoding and in behavioural performance (Supplementary Fig. 7). Noteworthy, however, at the level of condition averages, the patterns in target decoding and in behavioural reproduction accuracy showed excellent correspondence—both measures showed better performance for cued trials, larger cueing effects for distractor-present trials, prominent interference by the presentation of distractors and the largest interference for the earliest distractors.

## Discussion

Combining multivariate decoding and high temporal resolution EEG enabled us to investigate how anticipation influences the amount of sensory information extracted by the brain from target stimuli and temporally adjacent competing distractors. We observed two complementary effects—enhanced target identity coding and delayed interference from temporally adjacent distractors. Enhanced target processing and distractor resistance were furthermore each correlated with alpha oscillatory markers of preparatory attention, thus linking these target decoding effects to two independent operationalisations (cueing and variability in neural dynamics) of 'anticipatory state'. These effects emerged from a larger set of possible mechanisms by which anticipatory states may help resolve competition between sensory inputs that compete in time.

The relevant 'coding variable' for perception is not carried by response amplitude, but instead by stimulus identity information contained in these responses[38]. Previous fMRI work has already demonstrated that anticipatory expectations about features defining target identity can increase representational information in human visual cortex[39]. Our results show that even simple anticipation of stimulus timing, with no expectation that enables any feature-related template to be established, also significantly boosts target representations. Furthermore, by resorting to high temporal resolution EEG measurements, our results reveal that this occurs already during early sensory processing stages. Specifically, this 'representational boost' peaked around the classical N1 time range. Interestingly, however, while we also observed a parallel cueing effect on ERP amplitude (an amplified N1 response), the magnitude of the cueing effects on target identity decoding and on ERP N1 amplitude were uncorrelated.

This suggests that the influence of anticipatory cues on ERP amplitudes and on target identity decoding are mediated by complementary aspects of the EEG, and that the boost in target decoding cannot be simply attributed to a boost in response amplitude.

An open question remains what physiological mechanisms may underlie the observed enhancement in target decoding. As likely sources for this modulation, we consider a combination of heightened level of arousal, anticipatory orienting in time[2,3] and preparatory upregulation of neuronal populations coding for the relevant feature dimension (i.e., orientation channels). We speculate that each of these possible 'causes' may in turn be mediated by upregulation of the cholinergic system[40] (as well as possibly the norepinephrinic and dopaminergic systems[41]), in line with the observation that basal forebrain stimulation similarly enhances discriminability of visual input in rodents[42]. In the latter work, increased discriminability of visual responses was furthermore linked with decorrelation of neuronal firing rates in visual cortex. It is conceivable that macroscopic states of attenuated alpha oscillations (i.e., alpha 'desynchronization'[43]) provide a non-invasive index of such decorrelated visual activity[44].

In addition to a direct influence of anticipatory cues on target processing, we also observed a second effect that depended on distractor presence. While distractors always interfered with target decoding, this interference was delayed when targets could be anticipated. Anticipation may therefore enable adaptive perception by allocating a 'protective temporal window' from distractor interference, thereby possibly extending the high fidelity processing of the task-relevant target information and further orthogonalising target and distractor representations. Although speculative, similar protective windows may also contribute to phenomena such as the 'attentional blink'[45]. Interestingly, this delayed interference on target decoding by distractors occurred despite the fact that distractor decoding and distractor ERPs themselves appeared not to be delayed. How these observations are to be reconciled remains an important question for future research. One possibility is that, following anticipatory cues, distractor input is being routed to neural populations that show less overlap with those processing targets (despite the fact that both targets and distractors always occupied the same part of visual space). Another possibility is that this reflects increased investment in target processing only until sufficient target orientation information is extracted (after which distractor interference is 'tolerated' again). In future work, it will be interesting to evaluate whether the extent of the delayed interference varies with the amount of time required for perceptual processing; becoming shorter for easier tasks and longer for harder tasks.

Enhanced target decoding (across participants) and distractor resistance (across trials) were each also related to the attenuation of posterior alpha oscillations—an electrophysiological proxy for attentional engagement in human extracranial M/EEG measurements[28–33]. This was the case both for the task-related modulation by anticipatory cues (across participants), as well as for the spontaneous fluctuations in the absence of cues (across trials)—although variability in the cue-induced modulation across participants only correlated significantly with the main cueing effect on decoding, whereas the spontaneous variability across trials only correlated with the target decoding in distractor-present trials. Resolving this apparent discrepancy remains an interesting target for future research as it suggests there may be distinct sources of variability in posterior alpha oscillations that may have different bearings on perception. Still, by linking such states to the quality of content-specific early visual brain responses, the current work already makes an important extension to a growing

body of evidence suggesting a role for such states also in shaping response amplitudes[46] underlying neurophysiology[47,48], and perceptual[33,49], as well as mnemonic[50,51] performance.

A recent study also evaluated target decoding in the presence of distractors that were presented during a working memory delay[52] (for related behavioural studies see also ref. [53,54]). This study nicely demonstrated that the impact of distractors may be different in different brain areas, in their case impairing decoding of mnemonic representations in visual areas, while leaving them largely preserved in parietal areas. In our data, all effects occurred in posterior sites where decoding also peaked. In future studies, it will be interesting to resolve the specific areas in which these different effects occur, as well as to compare distractor-dependent effects that occur during time frames of encoding (as in the current work) with those during subsequent mnemonic retention (as in ref. [52]).

To maximise sensitivity, we focused on a set up with simple (but highly effective) temporal warning cues and with large centrally presented stimuli. Because of this, we cannot be sure whether our effects are driven primarily by changes in arousal, voluntary orienting of temporal attention or both. Still, by linking increased target decoding and distractor resistance observed with cueing also to states of attenuated posterior alpha oscillations, these data do provide a direct link to the voluntary attention literature where such brain states are commonly observed. Moreover, it should also be noted that the influence of the cues appeared largely specific to the targets and was particularly pronounced in distractor-present trials. This shows that the cues did more than merely boost all sensory information in a non-selective way. Rather, they specifically helped separating targets from distractors in time (an 'attentional' function). In future studies, it will be interesting to also track target and distractor identities in relation to more refined attentional and stimulus manipulations (e.g., embedding targets in streams of distractors, cueing different foreperiods, manipulating also spatial and feature-based expectations, etc.). Indeed, as this work showcases, high temporal resolution M/EEG stimulus identity decoding provides a powerful tool for reaching out to previously inaccessible questions regarding cognitive influences on sensory processing in humans (for similar arguments see ref. [14,15,55,56]). As this work highlights, this will prove particularly advantageous in tasks with rapidly changing displays as the decoded output appears largely robust against additive responses (unlike classical ERP responses; compare Fig. 4a with Fig. 5a) while maintaining excellent temporal resolution (unlike fMRI responses).

## Methods

**Participants**. Thirty healthy human volunteers (10 female; age range 19–35; mean age 25.5 years) participated in the study. This sample size was chosen based on a prior study that used the same decoding methodology and that yielded robust group-level conditional differences[22]. All participants had normal or corrected-to-normal vision and either held a university degree or were enrolled in university at time of participation. One participant was left handed. Data from all participants were retained for analysis. All participants provided written informed consent prior to participation and were reimbursed £10/h All experimental procedures were reviewed and approved by the Central University Research Ethics Committee of the University of Oxford.

**Stimuli, procedure and task**. Participants were seated in front of a monitor (100-Hz refresh rate) at a viewing distance of approximately 90 cm. We presented both visual and auditory stimuli (Fig. 1a). Visual grating stimuli consisted of six square wave cycles with a total diameter of 18 cm (11.4 degrees visual angle) such that the spatial frequency was approximately 0.53 cycles per degree. We randomly interleaved two types of gratings that were in anti-phase (gratings were either black or white centred), and over which we collapsed in all analyses. Grating orientations were randomly drawn, but were redrawn if within ±5 degrees from cardinal (0, 90, 180 degrees). We used the same stimuli for target, distractor and probe displays (see Fig. 1a), although their orientations were independently drawn. Distractors were presented in half the trials and were defined simply by their serial position

(i.e., the second grating). Distractors thus acted as visual masks, with the main difference with conventional masks being that distractors consisted of oriented gratings too, enabling us to decode and track both target and distractors identities. Targets and distractors were always presented for 50 ms each, and separated by an ISI of 20, 100 or 200 ms (on distractor-present trials), corresponding to a stimulus-onset-asynchrony (SOA) of, respectively, 70, 150 and 250 ms. Based on a prior pilot, we anticipated that the 100-ms ISI would yield the largest cueing benefit and we therefore used this ISI in the majority (80%) of distractor-present trials (Fig. 1a). Probe displays always appeared 500 ms after target offset (to avoid response-related contamination of the EEG traces immediately following target onset) and remained on the screen until the participant completed their orientation dial-up using the mouse (or until dial-up time ran out, see below). Auditory cues occurred in half the trials and consisted of 500-Hz pure tones that were presented for 50 ms. Cues indicated that the target would occur after 500 ms, but did not predict whether a distractor would also be present (i.e., cue and distractor presence were manipulated orthogonally). Inter-trial intervals (ITIs), defined as the interval between the response and the next target, did not differ between cue present and absent trials. To maximise the effect of the cues, ITIs were drawn from a truncated negative exponential distribution ranging between 600 and 5000 ms, with a mean of 1000 ms. Because this distribution approximates a flat hazard rate, target onset times were hard to predict, unless a cue was presented.

Participants' task was to reproduce the perceived orientation of the target stimulus as accurately as possible. To probe perception, we placed a probe grating on the screen, at a randomly drawn orientation. Participants used the computer's mouse to dial-up the perceived target orientation and clicked once satisfied. Participants were given unlimited time to decide what to report once the probe display appeared, but had to complete their dial-up within 2500 ms once they initiated their response. At response completion, feedback was provided by turning the fixation cross green for 300 ms for responses closer than ±15 degrees from the actual target orientation (with brighter green colours for more accurate responses); and red otherwise. In total, participants completed 30 blocks of 40 trials each, lasting about 1 h.

**Behavioural performance analysis**. We analysed both accuracy and RT. Accuracy was quantified as the absolute angular deviation between target orientation and reported orientation, while RT was quantified as the time from probe-onset to the first movement of the mouse to start the dial-up. We also analysed our circular reproduction responses using Bays' three-parameter mixture model[21] that models both the precision of the reports, as well as the respective proportions of target, distractor and guess responses. Finally, we obtained densities of responses relative to both target and distractor orientations. For this, we simply quantified the proportion of responses that were at a particular angular distance from either the target or the distractor. We evaluated response density in bins of ten degrees that we advanced from −90 to 90 degrees in steps of one degree.

**EEG acquisition and analysis**. EEG was acquired with Synamps amplifiers and Neuroscan acquisition software (Compumedics Neuroscan, North Carolina, USA). We used a custom 38-channel set up, sampling all electrodes posterior to the midline from the international 10–10 system and the rest from the associated 10–20 system; thus providing highest density at posterior sites of interest. Data were referenced to the left mastoid during recording, and re-referenced to an average-mastoid reference offline. The ground was placed on the left upper arm. Two bipolar electrode pairs recorded EOG. One pair was placed above and below the left eye (vertical EOG), whereas the other pair was placed lateral of each eye (horizontal EOG). During acquisition, data were band-pass filtered between 0.1 and 200 Hz, digitised at 1000 Hz and stored for offline analysis. All analyses were run on data with a sampling rate of 1000 Hz. Participant-specific trial-averaged ERP and decoding time courses were subsequently smoothed with a Gaussian kernel with a 15 ms standard deviation. This allowed us to bridge variability in the timing of the responses across participants, without smoothing away the essential characteristic of the ERP waveform (i.e., the distinct peaks). Such smoothing has similar consequences as low-pass filtering, which is also common in ERP research. We did confirm that our main results were largely invariant to this particular choice of smoothing (Supplementary Fig. 1).

Data were analysed in Matlab using a combination of FieldTrip[57] and custom code. During data preprocessing, we cut out our epochs of interest (relative to target onset), and removed excessively noisy epochs based on visual inspection of the signal's variance across trials and channels. Artefact rejection was performed on all trials, without knowledge of the conditions to which trials belonged. We additionally removed all trials in which targets and distractors may not have been perceived properly as a result of blinking. To this end, we iteratively removed all trials in which the vertical EOG contained samples with a z-score higher than five anywhere within the 400-ms window surrounding target onset. We did not explicitly cull for eye movements as the task was presented at fixation (although trials with large artefacts as a result of saccading would have likely been removed anyways based on our variance-based artefact rejection). After artefact rejection, there were 1042 ± 24 (mean ± 1 s.d.) trials left. Broken down by our main four conditions, numbers were: 291 ± 8 (cued, no distractor), 288 ± 7 (uncued, no distractor), 233 ± 5 (cued, distractor at 100-ms ISI) and 230 ± 7 (uncued, distractor at 100-ms ISI).

Time-frequency analysis was based on a short-time Fourier transform of Hanning tapered data. We estimated frequencies between 2 and 50 Hz in 1-Hz steps, using a 400-ms sliding time window that was advanced over the data in 80-ms steps. For relating pre-target alpha power to decoding, we also estimated alpha amplitude in a 500 ms pre-target window. Based on previous results using similarly short windows[51], we decided to use a relatively broad alpha range between 8–14 Hz. This also enabled us to use a multi-taper method[58] to obtain reliable single-trial estimates.

**EEG orientation decoding**. Stimulus orientation decoding was based on the broadband time domain signal that was preprocessed in two ways. First, a 250-ms pre-target trial-specific baseline was subtracted. We chose to position our baseline in the pre-target period because this interval is closest in time to the data period of interest (the target/distractor processing period), whilst not in itself containing any information regarding target/distractor identity. We did, however, confirm that highly similar results were obtained when positioning the baseline pre-cue, or when changing the duration of the baseline or subtracting the median as opposed to the mean baseline value from each trial (Supplementary Fig. 2). Second, the classical alpha band was filtered out of this signal. This was done to ensure that conditional differences in decoding could not be attributed to conditional difference in the signal's variance related to conditional differences in alpha amplitude (that we anticipated and observed). This is particularly relevant for interpreting the observed correlations of alpha amplitude (across trials) and amplitude modulation (across participants) with target orientation decoding. For filtering, we used an 8–14 Hz band stop Butterworth filter (two pass, filter order 4).

Visual orientation decoding was based on the multivariate (across electrode) Mahalanobis distance metric[22,59] using the data from the eight most posterior electrodes that showed the largest evoked response (O1, Oz, O2, PO7, PO3, POz, PO4, PO8). For generalisation, we applied a leave-one-out procedure. Because this procedure provides a trial-wise decoding estimate, this also enabled subsequent trial-wise correlation analyses with pre-target alpha amplitude and behavioural performance. Per trial, we calculated the Mahalanobis distance between that trial (the test trial) and all other trials (the reference or training trials) in which the target orientation was at a particular angular difference from the test trial. We did this for 19 bins of reference trials centred at angular differences ranging from −90 to +90 degrees (i.e., in steps of 10 degrees). For each bin (i.e., each orientation wedge), we included reference trials within ±22.5 degrees of the bin's centre. While we thus allowed substantial overlap between our bins (yielding smoother tuning profiles), we confirmed that highly similar results were obtained when not allowing any overlap between reference-bins. We then mean-centred the resulting distances across all angular bins and averaged the outcome across all trials within each of the experimental conditions. We ran this analysis separately for each time point, thus resulting in a time-resolved orientation tuning profile. For interpretability, we inverted this profile such that angular bins for which neuronal responses that were more similar to the test trial (and thus associated with a lower Mahalanobis distance) were associated with larger values. To capture orientation decoding in a single metric (per time point), we multiplied the mean-centred tuning profile with an inverted cosine function and averaged the result across all angular difference bins (as in ref. [22,23]). Due to low trial numbers, we did not consider the distractor-present trials with an ISI of 20 or 200 (these served primarily to demonstrate a temporally-specific effect of distractor timing on performance), except for one supplementary analysis presented in Supplementary Fig. 4. Target decoding incorporated all remaining trials in the reference ('training') set, whereas distractor decoding incorporated all remaining distractor-present trials.

**Statistical analysis**. Behavioural accuracy, RT and each of the mixture modelling outputs were compared between conditions using conventional repeated-measures analysis of variance, combined with paired samples t-tests.

Decoding time courses were statistically compared between conditions using cluster-based permutation tests[24] that effectively deal with the multiple comparisons encountered along the time axes. Specifically, this approach clusters neighbouring samples that survive univariate statistical testing ($p < 0.05$, two-tailed) and evaluates these clusters under a single permutation distribution of the largest cluster that is observed after permuting conditions (at the level of participant specific condition averages). We used 1000 permutations and considered both positive and negative clusters. For target decoding, we evaluated the main effects of cue presence and distractor presence, as well as their interaction (defined as cue present vs. absent for distractor-present vs. absent trials). For distractor decoding, we could only quantify the effect of cue presence.

In a complementary analysis, we also evaluated latency differences in the decoding time courses, on the basis of a Jackknife approach (as described in ref. [27]). Latency differences were estimated as the temporal difference (between cued and uncued conditions) at which the distractor interference effect first crossed the value associated with 50% of the maximal interference effect (the latter being estimated on the basis of the average of the cued and uncued interference effects). To obtain a Jackknife estimate of the reliability of the observed latency difference, we iteratively removed one participant from the participant pool and compared the resulting latency difference to the one observed when all participants were included. The Jackknife based estimate of the standard error then allowed us to compare the

observed latency difference against zero (i.e., the null hypothesis of no latency difference) under the student's t-distribution.

Correlations between target decoding and EEG amplitudes (across trials) or amplitude modulations (across participants) were quantified using Pearson's correlation coefficients. EEG amplitudes were averaged over all channels posterior to the midline where alpha amplitude, and its cue-related modulation, were most prominent. For the trial-wise correlation analysis, we partialled out trial number, as well as a trial-specific noise estimate that was anticipated to be associated with high amplitude (across most frequencies) and low decoding, thus constituting a potential confounding variable. This noise estimate was obtained by taking the trial-specific variance (across samples) of the high-pass filtered (40 Hz cut-off) data, and averaging this over all posterior electrodes. Correlations were again evaluated using cluster-based permutation analysis to circumvent the multiple comparisons encountered along the time and frequency axes.

All reported inferential statistics involved two-tailed tests, at an alpha level of 0.05.

**Data availability**. All data are publically available through the Dryad Data Repository at: https://doi.org/10.5061/dryad.fn664f0[60]. Code will be made available by the authors upon request.

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

## Acknowledgements

This research was funded by a Newton International Fellowship from The Royal Society and The British Academy (NF140330), as well as Marie Skłodowska-Curie Fellowship from the European Comission (ACCESS2WM) to F.v.E., and was supported by the NIHR Oxford Health Biomedical Research Centre, a Wellcome Trust Senior Investigator Award (104571/Z/14/Z) and a James S. McDonnell Foundation Understanding Human Cognition Collaborative Award (220020448) to A.C.N., and a Medical Research Council Career Development Award (MR/J009024/1) to M.G.S. The Wellcome Centre for Integrative Neuroimaging is supported by core funding from the Wellcome Trust (203139/Z/16/Z). The views expressed are those of the authors and not necessarily those of the National Health Service, the National Institute for Health Research or the Department of Health. We also wish to thank Marcel Niklaus and Nick Myers for their input on the experimental design and their assistance during data collection and analysis, and Alex Board for his help with the bibliography.

## Author contributions

F.v.E., M.G.S. and A.C.N. designed the study and wrote the paper. F.v.E. and S.R.C. conducted the experiments. F.v.E., S.R.C. and M.G.S. analysed the data.
