## [Peer Review File · Nature Communications]

Reviewers' comments:

Reviewer #1 (Remarks to the Author):

The authors seek to identify how temporally-oriented attention alters neural representations of visual stimuli that are relevant to behavior (targets) and those that are irrelevant (distractors). They apply a rigorous EEG-based decoding approach to quantify the fidelity of neural representations at a millisecond timescale, and through several clever analyses, establish not only that target representations are enhanced when a cue is presented, but also that the interfering effects of irrelevant distractor stimuli on EEG decoding are delayed. Furthermore, the authors evaluated whether anticipatory alpha oscillations are related to the fidelity of neural representations. Indeed, on trials with attenuated alpha power at posterior recording sites, target representations were enhanced and distractor interference was reduced. The authors conclude that anticipatory state, either altered via a temporal attention cue or via endogenous changes in posterior alpha power, can alter processing of relevant visual targets, even when distracting visual stimuli are presented.

Altogether, I think this manuscript will make a really excellent contribution to Nature Communications. I suggest several additional analyses that would further expand its reach, especially by evaluating the relationship between neural representations and behavioral performance. While I encourage the authors to attempt these analyses and report results whichever way they turn out, the manuscript stands quite strongly as-is, and negative results from such analyses will not adversely alter my enthusiasm for this report. I also have several questions about the analyses & results, which the authors may be able to address via minor revisions of the text and figures.

Major

1. Can the authors relate aspects of neural target/distractor processing/representation to behavioral performance on the task?

2. Related to (1), behavioral performance should be more comprehensively analyzed/reported. This appears to be an orientation recall task, like is commonly used with visual WM experiments. There is a recent literature on effects of visual distraction on visual WM representations and behavioral performance (see Rademaker et al, 2015; Bettencourt & Xu, 2015, for examples). What kinds of effect(s) does the visual distractor stimulus have on the behavioral report? Are response error distributions biased in the direction of the distractor? Are they less precise? In general, I encourage the authors to report parameters of a mixture model (e.g., Bays, Zhang & Luck, or similar model) fit to error histograms, and discuss whether or not they see evidence for distractor-induced biases and/or broadening of response distributions. This will also help clarify the source of the increased errors with distractors presently plotted in Fig. 1b. Perhaps one or another of these mixture model parameters will be related to neural measures of target/distractor representations (point 1 above).

3. For all decoding analyses, it seems as though alpha-filtered data were used (after removing all signal from the 8-14 Hz alpha band). Is it possible to decode orientation representations during the post-target/distractor period using this alpha signal alone? I'm mainly thinking of the recent work of Joshua Foster & Ed Awh. While they use spatial tasks, it could be that subjects are using a spatial strategy here for the delayed response task, which is quite similar to the WM tasks they use. It would be especially interesting to see whether the distractor alters the fidelity of the alpha-band representation in a similar manner to how it alters the broad-band (minus alpha) representation, as reported in Fig. 3. In any case, some discussion of how these results fit into that literature may be useful.

4. The alpha range chosen for the analyses reported in Fig. 4 is quite broad – is it possible to limit this analysis to each individual participant's peak alpha frequency (if available)? Or, instead, discuss the potential advantages/limitations of using a larger frequency band.

5. Is there a latency difference in the distractor interference effect (as in Fig. 3b) as a function of pre-trial alpha (perhaps within uncued trials)? It appears as though there is in Fig 4c, but a similar plot to Fig. 3b may be useful to readers.

6. Is it possible to demonstrate whether the 'protective window' (distractor interference latency) is utilized by participants? Perhaps the authors could find a way to split trials based on long vs short-latency distractor interference effects and compare behavioral performance? Such a scheme may be challenging, and perhaps this remains a question best addressed by future experiments. But I would be interested to hear the authors thoughts on this (related to points 1 & 2 above)

7. On pg 14, the methods which describe trial binning are somewhat confusing. Specifically, this sentence is challenging to interpret: "For each bin, we included training trials whose angular difference from the test trial were within ± 22.5 degrees of the bin's center". First, what does the angular difference from the test trial have to do with the training bins? Second (and perhaps I've misinterpreted), does this mean that, for each bin used for training, all trials within a 45-deg wide feature wedge (1/4 of feature space) were used? If so, does this mean a given training trial counted multiple times (as the bins were spaced by 10 deg it seems)? That seems odd to me – some justification of this choice, or a demonstration that using exclusive/non-overlapping boundaries for each bin would help clarify, if this is a correct interpretation.

8. How will the data/code be shared?

Minor

1. In the abstract, there's a small typo: "but, instead, delay[s] it"

2. Fig. 1c – is it possible to break out the overlaid 'fidelity' plot (which is on top of the pattern similarity image) into a separate panel? This panel is especially dense at present, and readers may benefit from smaller chunks of digestible data. Additionally, adding a color bar for panel 1c would also help readers interpret the data.

3. Fig. 1c – can this analysis be shown separately for target representations (on no-distractor trials) and distractor representations (on distractor+ trials)?

4. Fig. 4a-b – again, very dense figure panels. If possible, separating the topographies and scatterplot from the TFR would make these figures easier to digest, and features of the data easier to see. This would also avoid the problem of covering up portions of panel a with the topographies.

5. Are trials culled based on eye movements? The methods only describe removing trials with identified blink artifacts

Reviewer #2 (Remarks to the Author):

Using EEG decoding methods in a visual orientation reproduction task with preparatory auditory cues and visual distractors, the manuscript shows that the representations of target and distractor orientations in broad-band time-domain EEG signals (ERPs) are affected by the preparatory cue in two

specific ways: (1) enhancing the representation of the target stimulus; and (2) delaying the interference of distractors with target representations. These two changes in EEG target decoding by the auditory cue are then related to the attenuation of posterior alpha oscillations. The authors conclude that anticipation is mediated by alpha oscillations through target representation enhancement and delayed distractor interference.

In my view, the data provided in this manuscript is insufficient to advance this conclusion. I have three main objections:

1) The preparatory cue is shown to improve behavioral performance in this task, but none of the brain signal analyses are then shown to be directly related to this behavioral effect. The EEG differences observed between cued and uncued trials could be related to multisensory processing, divided attention, and not specifically to anticipating visual perception. This is particularly disappointing given the fact that the decoding methods used can power single-trial analyses (see Fig. 4d) and thus relate to behavioral errors even on a trial-by-trial basis (see for instance Kok et al. 2012). Such approaches could be used at several points in the manuscript to validate the interpretations on the relation of EEG signals with behavior. For instance, the 2 mechanisms identified in Fig. 3 could be specifically linked to reproduction error by restricting the analysis to uncued trials and checking correlation between cue decoding and reproduction error, and distractor interference latency and reproduction error. Similarly, the relationship of alpha power with decoding identified in Fig. 4 should also correlate with reproduction error to validate the current interpretation of the manuscript. The behavioral impact of the visual mask is also known to be different depending on the similarity of target and distractor (Magnussen et al. 1991; Magnussen and Greenlee 1992, 1999; Rademaker et al. 2015), this could also be validated in their behavioral data and then used to test if distractor orientation difference affects target decoding in a direction consistent with the behavioral effects. Also the ISI dependence of the behavioral effect in Fig. 1b could be used to test the relationship of target decoding effects and behavior. An additional component of their task that could provide behavioral parameters to relate with the subjects' anticipatory state is the ITI. Despite the flat hazard rate defined for the ITI, it is likely that there is a subjective urgency that can have behavioral impact (Janssen and Shadlen, 2005) and impact correspondingly ERP decoder modulations.

2) The association of alpha-band activity with the two EEG mechanisms (ERP target decoding accuracy and distractor interference latency) is sketchy and not consistent. Figure 4c-d show nicely how pre-stim alpha power acts very similarly to the anticipatory cue, in relation to the distractor interference effect of Fig. 3. However, these same panels show that pre-stim alpha does not reproduce the main effect of the auditory cue (main effect of cue in Fig. 3c). This is in contrast with the message of Fig. 4b. Are there 2 alphas? A cue-induced alpha that is responsible for the main effect of cue, and a spontaneous alpha that accounts for the interaction effect (distractor interference)? While the spontaneous alpha analysis in Fig. 4c-d is very suggestive, the interaction effect reported in Fig. 3 is triggered by the cue presentation, so it should also be present in the analysis of cue-induced alpha. Does the cue-induced alpha correlate in an interindividual analysis with the magnitude of the interaction effect? In Discussion it is suggested that both spontaneous and cue-induced alpha correlate with the main effect and the interaction effect ("This was the case both for the task-related modulation..."), but this is currently not supported by Fig. 4. An additional aspect of alpha-band activity that is not addressed in this analysis is the role of alpha phase. Recent studies are showing that alpha paces the sequence of perceptual cycles (VanRullen 2016) and that auditory stimuli reset the posterior alpha rhythm, with perceptual consequences (Romei et al., 2012). The timings and intervals of interest in the current task are within one alpha cycle, possibly making the effects sensitive to alpha phase. Could a sizable part of the behavior and target decoding results be explained by the alpha phase-resetting of the auditory cue?

3) out of the 2 mechanisms identified in this manuscript, the main effect of the cue has already been reported before (Kok et al 2012). The manuscript does not discuss this influential, recent result at all.

In addition I have a number of methodological concerns:

1) "Participant-specific trial-averaged ERP and decoding time courses were subsequently smoothed with a Gaussian kernel with a 15 ms standard deviation". I miss a strong argument to perform such operation in the final average. This looks like a "cosmetic" preprocessing step and "it obscures the ability to evaluate the physiological plausibility of an effect, and thereby hides relevant complementary information from readers" (Van Ede & Maris 2016).

2) "Significant clusters". Wrong concept that appears repeatedly in the manuscript (Figure 3 and Figure S1 captions; "three significant clusters", page 4, etc). The non-parametric cluster-based permutation test serves to test a null hypothesis: The data (not the parameters estimated from the data) in different experimental conditions came from the same probability distribution, so they are exchangeable. The alternative hypothesis consists that the data in different experimental condition do NOT come from the same probability distribution. Stating "there is a significant cluster" is simply wrong. The statistical significance indicates an informed decision about the uncertainty to accept or reject the null hypothesis but never about "when" (time) or "where" (topography, frequency) those differences take place. The correct statistical conclusion would be that the authors have found a significant difference between condition A v.s. condition B. I encourage the authors to revisit Maris & Oostenveld (2007) and Maris (2012). Check:

http://www.fieldtriptoolbox.org/faq/how_not_to_interpret_results_from_a_cluster-based_permutation_test

3) EEG orientation decoding (in Methods). The authors state that "a 250 ms pre-target baseline was subtracted". Was it performed at a single-trial level? Such a short period will contain a lot of noise and a demeaning operation (mean subtraction of the entire epoch) would be way more efficient. If data is stationary, single-trial baseline correction would correspond to baseline correction averaging the data across trials. Please check the stationarity of your data: do you find significant differences in your ERP and decoding plot subtracting the single trial baseline estimates v.s. the grand-mean baseline subtraction? If the answer is positive, please consider a more careful data normalization (see Grandchamp & Delorme 2011).

Minor comments:

Please provide the number of trials per condition left after artifact rejection. This is important information to interpret the decoding results and to design future experiments based on your findings. As much as possible, authors should follow well-established guidelines (Keil et al., 2014).

Figure 1c: There is no color bar associated. Please add the proper color bar to the figure or specify in the figure legend that Figure 1c and d share the same color bar.

Figure 1c summary decoding statistic is not clear. The y-scale is difficult to read and this statistic is the one employed throughout the paper. I think it deserves a subfigure in its own right. Figure 2bc in Wolf et al., (2017) could be a good choice.

Figure 3a: the legend is confusing and it is not well explained. For example, the black line "distr –"

never appears in the figure. Same goes to Figure S1a. Please clarify this because takes time to understand your key figures.

Figure 4b,c inset plots are too small.

References:

- Grandchamp, R., & Delorme, A. (2011). Single-trial normalization for event-related spectral decomposition reduces sensitivity to noisy trials. *Frontiers in psychology*, 2.
- Keil, A., Debener, S., Gratton, G., Junghöfer, M., Kappenman, E. S., Luck, S. J., ... & Yee, C. M. (2014). Committee report: publication guidelines and recommendations for studies using electroencephalography and magnetoencephalography. *Psychophysiology*, 51(1), 1-21.
- Grandchamp, R., & Delorme, A. (2011). Single-trial normalization for event-related spectral decomposition reduces sensitivity to noisy trials. *Frontiers in psychology*, 2.
- Kok, P., Jehee, J. F., & De Lange, F. P. (2012). Less is more: expectation sharpens representations in the primary visual cortex. *Neuron*, 75(2), 265-270.
- Magnussen, S., & Greenlee, M. W. (1992). Retention and disruption of motion information in visual short-term memory. *Journal of Experimental Psychology: Learning, Memory, and Cognition*, 18, 151–156.
- Magnussen, S., & Greenlee, M. W. (1999). The psychophysics of perceptual memory. *Psychological Research*, 62, 81–92.
- Magnussen, S., Greenlee, M. W., Asplund, R., & Dyrnes, S. (1991). Stimulus-specific mechanisms of visual short-term memory. *Vision Research*, 31, 1213–1219.
- Maris, E. (2012). Statistical testing in electrophysiological studies. *Psychophysiology*, 49(4), 549-565.
- Maris, E., & Oostenveld, R. (2007). Nonparametric statistical testing of EEG-and MEG-data. *Journal of neuroscience methods*, 164(1), 177-190.
- Rademaker, R.L., Bloem, I.M., De Weerd, P., & Sack, A. T. (2015) The impact of interference on short-term memory for visual orientation. *Journal of Experimental Psychology: Human Perception and Performance*, 41(6), 1650-1665
- Romei, V., Gross, J., & Thut, G. (2012). Sounds reset rhythms of visual cortex and corresponding human visual perception. *Current Biology*, 22(9), 807-813.
- van Ede, F., & Maris, E. (2016). Physiological plausibility can increase reproducibility in cognitive neuroscience. *Trends in Cognitive Sciences*, 20(8), 567-569.
- VanRullen, R. (2016). Perceptual cycles. *Trends in Cognitive Sciences*, 20(10), 723-735.
- Wolff, M. J., Jochim, J., Akyürek, E. G., & Stokes, M. G. (2017). Dynamic hidden states underlying working-memory-guided behavior. *Nature Neuroscience*, 20(6), 864-871.

Dear Editors, Dear Reviewers,

We were very pleased to receive such overall enthusiasm for our manuscript, as well as the large number of constructive comments and suggestions for further analyses. Having embraced this valuable feedback, we are now pleased to resubmit a considerably improved and much more thorough and comprehensive manuscript.

For your convenience, we also summarise the most important revisions below:

1. We have included a new complementary analysis that further strengthens our conclusion that preparatory cues facilitate target processing in the face of temporal distractors by reducing their immediate interference. We outline these results below, before turning to our point-by-point replies.
2. To better convey the key features of the behavioural data and of the applied decoding approach, we have separated previous Figure 1 (task, performance, and decoding) into two separate figures. For of the behavioral data (updated Figure 1), we have now added the results from a mixture-modelling analysis revealing widespread effects of the cues on performance. To better emphasize the utility of our decoding approach (updated Figure 2), we now first showcase that we can individuate and track in time both target and distractor representations, even when these are presented in close temporal proximity.
3. We have now embedded several relevant references brought forward by both reviewers.
4. We have modified how we report our cluster-based permutations statistics to adhere to recommended guidelines.
5. We now explicitly address the correspondence between target decoding and behavioral performance.
6. We have moved the previous the Supplementary Figure showing the corresponding ERP results into the main body, recognising that the comparison between decoding and ERP results is relevant for several key points in our manuscript.
7. We have added a Supplementary Figure demonstrating that our main effect of interest are invariant to the chosen amount of data smoothing.

We are again very grateful for your time in considering our manuscript.

Yours faithfully,

Freek van Ede, Sammi Chekroud, Mark Stokes, Kia Nobre

Before turning to our point-by-point replies

Our central aim was to investigate how preparatory cues prioritise target representations relative to temporally competing distractor stimuli. We have now included the outcomes of a new, complementary, analysis which we feel provides further direct support for our conclusions. We introduce and explain this new analysis before turning to our point-by-point replies. As you will see below, this analysis provides converging evidence supporting our interpretation that cues can help overcome temporally competing distractors by reducing their immediate interference.

Specifically, we have now added the following to our Results and Supplemental Information (please note that the term “reference set” is now used to refer to what we previously termed “training set”, following a comment from reviewer 1):

p7/8 (Results): “[...] The pattern of results suggested that cues are particularly useful for protecting target analysis from distraction. To test this possibility more directly, we performed an additional, complementary analysis. We reasoned that, if cues protected target analysis from interference, then target representations in distractor-present trials with a cue should more closely resemble distractor-absent trials than should distractor-present trials without a cue (i.e., in distractor-present trials, cues should make the target representation appear more as if there was no distractor). To test this prediction, we re-evaluated target orientation decoding, but this time only included distractor-absent trials (irrespective of cue condition) in our reference (“training”) set. This confirmed that, indeed, immediately after distractor onset, the target orientation could be better decoded from cued compared to uncued distractor-present trials (Fig. S2).

Figure S2. Target decoding in distractor-present trials, when only including distractor-absent trials in the reference set. (a) Orientation decoding tuning profiles separately for cued and uncued trials, as well as their difference. Reference sets were the same for cued and uncued trials; the reference set simply contained all distractor-absent trials, independent of cue-presence. (b) Corresponding time-resolved summary statistics of target decoding as a function of cue condition. Shadings represent ± 1 s.e.m. calculated across participants ($n = 30$).

Point-by-point replies

Reviewer 1

The authors seek to identify how temporally-oriented attention alters neural representations of visual stimuli that are relevant to behavior (targets) and those that are irrelevant (distractors). They apply a rigorous EEG-based decoding approach to quantify the fidelity of neural representations at a millisecond timescale, and through several clever analyses, establish not only that target representations are enhanced when a cue is presented, but also that the interfering effects of irrelevant distractor stimuli on EEG decoding are delayed. Furthermore, the authors evaluated whether anticipatory alpha oscillations are related to the fidelity of neural representations. Indeed, on trials with attenuated alpha power at posterior recording sites, target representations were enhanced and distractor interference was reduced. The authors conclude that anticipatory state, either altered via a temporal attention cue or via endogenous changes in posterior alpha power, can alter processing of relevant visual targets, even when distracting visual stimuli are presented.

Altogether, I think this manuscript will make a really excellent contribution to Nature Communications. I suggest several additional analyses that would further expand its reach, especially by evaluating the relationship between neural representations and behavioral performance. While I encourage the authors to attempt these analyses and report results whichever way they turn out, the manuscript stands quite strongly as-is, and negative results from such analyses will not adversely alter my enthusiasm for this report. I also have several questions about the analyses & results, which the authors may be able to address via minor revisions of the text and figures.

Thank you. We agree that it is useful to include the proposed additional analyses, irrespective of their outcomes. As you will see, we have now added the outcomes of these analyses and have made the requested revisions to our text and figures.

Major

1. Can the authors relate aspects of neural target/distractor processing/representation to behavioral performance on the task?

Unfortunately, our task parameters were such that they may not have enabled us to derive reliable correlations between trial-by-trial variability in target decoding and trial-by-trial variability in behavioural performance. Many psychological functions likely intervened between perceptual decoding and the final response made, possibly hampering our ability to find direct correlations between these measures. Examples being post-target lapses in short-term memory, post-target changes of mind as to which of the items was the target, perceptual interference by the probe stimulus, motoric errors, and so on. Nevertheless, we agree with the reviewer that it is important to disseminate the results of these correlational analyses, irrespective of their outcomes, we have therefore now added the following to our manuscript – also discussing why we might not have observed any correlations and emphasizing that the patterns of target decoding and behavioural accuracy did correspond really well across conditions:

p11 (Results): ***“Correspondence between decoding and behavioural performance***

In a simple perceptual task one would expect the quality of target decoding to correlate with behavioural performance on a trial-by-trial basis. In our task, however, many factors are likely to influence behavioral performance, of which the perceptual processing that takes place during initial encoding is but one. Indeed, behavioural performance on single trials is likely influenced by many additional factors that are not well captured by the early EEG responses that we focused on (such as post-target lapses in short-term memory, post-target changes of mind as to which of the items was the target, perceptual interference by the probe stimulus, motoric errors, and so on.). This may explain why we were not able to demonstrate compelling and consistent correlations between the trial-by-trial variability in the magnitude of target decoding and in behavioural

performance (Fig. S6). Furthermore, we note that the experiment was designed to compare decoding and performance among conditions, and not for maximising variability within the conditions. This will have further compromised the sensitivity for such correlations. However, it should at the same time be noted that, at the level of condition averages, the patterns in target decoding and in the behavioural reproduction accuracy showed excellent correspondence – both measures showed better performance for cued trials, larger cueing effects for distractor present trials, prominent interference by the presentation of distractors, and the largest interference for the earliest distractors.

Figure S6. Trial wise correlations between target decoding and behavioural performance. Time-resolved trial-wise correlations between target decoding and reproduction error (left) as well as reaction time (right), as a function of cue and distractor presence. Note that we hypothesised negative correlations, provided that better behavioural performance is associated with lower reproductions errors and lower RTs. The lines show the group-mean correlation values and the shadings represent ± 1 s.e.m. calculated across participants ($n = 30$).

2. Related to (1), behavioral performance should be more comprehensively analyzed/reported. This appears to be an orientation recall task, like is commonly used with visual WM experiments. There is a recent literature on effects of visual distraction on visual WM representations and behavioral performance (see Rademaker et al, 2015; Bettencourt & Xu, 2015, for examples). What kinds of effect(s) does the visual distractor stimulus have on the behavioral report? Are response error distributions biased in the direction of the distractor? Are they less precise? In general, I encourage the authors to report parameters of a mixture model (e.g., Bays, Zhang & Luck, or similar model) fit to error histograms, and discuss whether or not they see evidence for distractor-induced biases and/or broadening of response distributions. This will also help clarify the source of the increased errors with distractors presently plotted in Fig. 1b. Perhaps one or another of these mixture model parameters will be related to neural measures of target/distractor representations (point 1 above).

Thank you for this valuable suggestion. We have now subjected our behavioral data to the three-parameter mixture model by Bays et al. (2009), again including for distractor-present trials only those that occurred at the 100-ms ISI to ensure sufficient trials for analysis. As depicted in updated Figure 1 (also pasted below), this revealed a substantial proportion of so called ‘swapping errors’ (reporting the distractor orientation instead of the target orientation) that were reduced by the cue, thus showing that the cue was particularly helpful in separating the target from the distractor in time in order to judge which one came first (i.e., which item was the target). We also found that cues increased response precision and decreased guess rates, specifically in distractor-present trials. These additional results thus all converge on our interpretation that the cues are particularly beneficial for visual perception in face of temporal distraction. We have now updated Figure 1 as shown below, and have added the associated text to our manuscript that is also pasted below:

Redacted

Figure 1. Task design and behavioral performance. (a) Visual orientation reproduction task with preparatory auditory cues and visual distractors. Participants reproduced the orientation of the visual target grating using a computer mouse. In half the trials, targets were preceded by an auditory warning cue. Targets could be followed by no distractors, or by a visual distractor at one of three ISIs (20, 100, 200 ms). Target-probe intervals and inter-trial intervals were drawn independently of cue and distractor presence. (b) Average orientation reproduction errors (in degrees) and reaction times (in ms) for cued and uncued trials as a function of distractor presence and ISI. (c) Mixture-modelling parameters as a function of cue and distractor presence. c+ for cue present, d- for distractor absent, and so on. (d) Response distributions centred on the target and the distractor orientation. To ensure sufficient trial numbers, we only considered distractor-present trials in the 100-ms ISI condition in panels c and d. Error bars represent ± 1 s.e.m. calculated across participants ($n = 30$). ISI = inter-stimulus-interval; SOA = stimulus-onset-asynchrony. * $p < 0.05$; ** $p < 0.01$; *** $p < 0.001$.

p4 (Results): “To further interrogate the behavioral performance data, we also ran a mixture modelling analysis (Bays et al., 2009), quantifying the precision of the orientation reproduction reports, alongside the proportion of reports classified as a target report, a distractor report (‘swapping error’), or a random guess. Each panel in Figure 1c shows these parameters as a function of cue and distractor presence. For precision, we observed a significant main effect of

cue presence, with higher precision for cued compared to uncued trials ($F_{(1,29)} = 7.514, p = 0.01, \eta_p^2 = 0.206$), as well as a significant main effect of distractor presence, with lower precision for distractor-present trials ($F_{(1,29)} = 68.569, p = 3.959e^{-9}, \eta_p^2 = 0.703$). Although the interaction between cue presence and distractor presence was not significant ($F_{(1,29)} = 1.767, p = 0.194, \eta_p^2 = 0.057$), planned comparisons revealed that cues increased precision in distractor-present ($t_{(29)} = 3.598, p = 0.001, d = 0.657$), but not in distractor-absent trials ($t_{(29)} = -0.877, p = 0.388, d = 0.160$). In addition, we observed that cues increased the number of target reports in the distractor-present ($t_{(29)} = 4.688, p = 6.035e^{-5}, d = 0.856$), but not the distractor-absent trials ($t_{(29)} = -1.054, p = 0.301, d = -0.192$), this time also marked by a significant interaction between cue and distractor presence ($F_{(1,29)} = 27.619, p = 1.247e^{-5}, \eta_p^2 = 0.488$). Complementing this increase in target reports in distractor-present trials, we also found that cues reduced the number of distractor reports in these trials ($t_{(29)} = -4.360, p = 1.492e^{-4}, d = -0.796$). Finally, we observed a significant interaction between cue and distractor presence also for the proportion of guess reports ($F_{(1,29)} = 9.006, p = 0.006, \eta_p^2 = 0.237$), where cues significantly reduced the proportion of guesses in distractor-present ($t_{(29)} = -2.336, p = 0.0266, d = -0.426$) but not in distractor-absent trials ($t_{(29)} = 1.054, p = 0.301, d = 0.192$).

The impact of the cue on performance in our task is further visualized in Figure 1d, showing the respective response distributions aligned to the target and distractor orientations. This confirmed that, when there are no distractors (left panel), response distributions look very similar between cued and uncued trials. However, in face of temporal distractors (right panel), it becomes clear that cues increase the proportion of target responses (solid lines), while reducing the proportion of distractor responses (dashed lines). These data thus collectively reveal a key contribution of the cue in benefiting visual perception in face of temporal distractors, and helping to separate the occurrence of targets from distractors in time.”

Finally, note that we have also added a paragraph to our Discussion where we bring forward this relevant related work on distraction in visual working memory:

p13 (Discussion): “A recent study also evaluated target decoding in the presence of distractors that were presented during a working memory delay (Bettencourt and Xu, 2016; for related behavioral studies see also Magnussen and Greenlee, 1992; Rademaker et al., 2015). This study nicely demonstrated that the impact of distractors may be different in different brain areas, in their case impairing decoding of mnemonic representations in visual areas, while leaving them largely preserved in parietal areas. In our data, all effects occurred in posterior sites where decoding also peaked. In future studies, it will be interesting to resolve the specific areas in which these different effects occur (for example using MEG or iEEG, instead of EEG), as well as to compare distractor-dependent effects that occur during time frames of encoding (as in the current work) with those during subsequent mnemonic retention (as in Bettencourt and Xu, 2016).”

3. For all decoding analyses, it seems as though alpha-filtered data were used (after removing all signal from the 8-14 Hz alpha band). Is it possible to decode orientation representations during the post-target/distractor period using this alpha signal alone? I’m mainly thinking of the recent work of Joshua Foster & Ed Awh. While they use spatial tasks, it could be that subjects are using a spatial strategy here for the delayed response task, which is quite similar to the WM tasks they use. It would be especially interesting to see whether the distractor alters the fidelity of the alpha-band representation in a similar manner to how it alters the broad-band (minus alpha) representation, as reported in Fig 4.

Thank you for bringing this up, this is an interesting point. We removed the alpha-band to ensure that the observed increase in target decoding with the cue could not simply be attributed to reduced variance in the signal as a consequence of attenuated alpha oscillations. Still, we agree that it is also of interest to see whether alpha itself may also contain target-identity information in line with these recent studies. As suggested, we have now also analysed stimulus identity decoding for Hilbert-

transformed alpha-band amplitude envelope data (i.e., for topographical patterns of alpha-band amplitude). While this confirmed that it was indeed possible to decode stimulus orientation from alpha amplitudes, decoding was less robust than when using the broad-band signal and conditional differences were no longer apparent. Because we think others may be curious about this too, we have added the following to our Results section and Supplemental Information:

p10 (Results): “At the same time, several recent studies have shown that, at least in the context of working memory, not only target location (Foster et al., 2015) but also orientation (Fukuda et al., 2016) can also be decoded from the topographical pattern of alpha amplitudes. While we could confirm such alpha-based orientation decoding in our data, this appeared less robust and did not yield clear conditional differences as a function of cue and distractor presence (Fig. S5).”

Figure S5. Target and distractor decoding based on alpha amplitude envelopes. Conventions as in Figures 2a and 4a, except before running the decoding analysis, we band-pass filtered the time-domain signal in the 8-14 Hz alpha band and used the Hilbert transform to obtain time-varying amplitude envelopes. Shadings represent ± 1 s.e.m. calculated across participants ($n = 30$).

3. In any case, some discussion of how these results fit into that literature may be useful.

See above, we now bring this relevant literature forward in our manuscript.

4. The alpha range chosen for the analyses reported in Fig. 4 is quite broad – is it possible to limit this analysis to each individual participant’s peak alpha frequency (if available)? Or, instead, discuss the potential advantages/limitations of using a larger frequency band.

Because of the relatively short time window available for the frequency analysis (in our case, a 500 ms pre-target window), we only have limited frequency resolution to begin with. Opting for a “broad” alpha frequency range enabled us to use a multi-taper method for spectral estimation, which has high sensitivity for estimating amplitude in a particular frequency range (see Percival and Walden, 1993). We should also note that this “broad range” is where we not infrequently observe our effects, when analysed with similarly short time windows (as for example in van Ede et al., 2017). Prompted by the reviewer’s suggestion, we also tried estimated individuals’ peak frequencies and estimated amplitude

in a ± 2 and ± 3 Hz range around this peak. This yielded nearly identical results without any observable advantage, so we have not further followed-up on this. We have, however, better specified our motivation for this relatively broad range in our Methods:

p18 (Methods): “For relating pre-target alpha power to decoding, we also estimated alpha amplitude in a 500 ms pre-target window. Based on previous results using similarly short windows (van Ede et al., 2017), we decided to use a relatively broad alpha range between 8-14 Hz. This also enabled us to use a multi-taper method (Percival and Walden, 1993) to obtain reliable single-trial estimates.”

5. Is there a latency difference in the distractor interference effect (as in Fig. 3b) as a function of pre-trial alpha (perhaps within uncued trials)? It appears as though there is in Fig 4c, but a similar plot to Fig. 3b may be useful to readers.

Thank you for bringing this up. We agree that the equivalent plot is informative, and have therefore now included this in our updated Figure 6 (also to be found below a different comment regarding this Figure from reviewer 2). This revealed a qualitatively similar latency effect, although our Jackknife quantification reached significance only for the higher thresholds at which we quantified the latency shift. We must point out, however, that these results were based on a median-split of the data (i.e., of the pre-target alpha amplitudes) which is inevitably less sensitive than an analysis that takes into account the continuous variability in the data (a point that we now make clearer in the text). It is for this reason that we also included, and focused our main statistical analysis of this effect, on the trial-wise correlation between alpha amplitude and decoding (Fig. 6f).

6. Is it possible to demonstrate whether the ‘protective window’ (distractor interference latency) is utilized by participants? Perhaps the authors could find a way to split trials based on long vs short-latency distractor interference effects and compare behavioral performance? Such a scheme may be challenging, and perhaps this remains a question best addressed by future experiments. But I would be interested to hear the authors thoughts on this (related to points 1 & 2 above).

This is a really important point, but one that is indeed very challenging to address. For example, the distractor interference effect is quantified as a difference in target decoding between different trials (those with a distractor present and those without a distractor). As such, this effect (as quantified this way) simply does not exist within single trials. Moreover, as responded to above, trial wise correlations between decoding and behavioral performance did not yield compelling results either. That said, this comment did prompt us to further think about this point and to speculate on a possible extension of this result in future studies:

p12 (Discussion): “In future work, it will be interesting to evaluate whether the extent of the delayed interference may vary with the amount of time required for perceptual processing; becoming shorter for easier tasks and longer for harder tasks.”

7. On pg 14, the methods which describe trial binning are somewhat confusing. Specifically, this sentence is challenging to interpret: “For each bin, we included training trials whose angular difference from the test trial were within ± 22.5 degrees of the bin’s center”. First, what does the angular difference from the test trial have to do with the training bins?

To obtain tuning profiles, we always look at the multivariate Mahalanobis distance between a given test trial, and all remaining trials (what we had referred to as “training” trials) that have a certain angular difference from this particular test trial. In this way, we were able to generate a tuning profile of similarity to the test trial (centered at orientation 0) as a function of angular difference with the training trials (ranging from -90 to +90 degrees). This is different from other “decoding” approaches (such as the forward encoding modelling approach) in which training trials are used to obtain weights

that can then be used to reconstruct tuning in test trials. Upon reflection, we now believe the term “training trials” may not be technically appropriate for describing our approach and may therefore be potentially misleading. As a better term, we now refer at all relevant instances to “reference trials” and “reference set” instead of “training trials” and “training set”.

Having clarified this, we agree that this sentence was unnecessarily confusing. We have therefore simplified this to: “For each bin (i.e., each orientation wedge), we included reference trials within ± 22.5 degrees of the bin’s center.” which we anticipate will be much easier to parse in the context of the preceding sentences (page 19, Methods).

Second (and perhaps I’ve misinterpreted), does this mean that, for each bin used for training, all trials within a 45-deg wide feature wedge (1/4 of feature space) were used? If so, does this mean a given training trial counted multiple times (as the bins were spaced by 10 deg it seems)? That seems odd to me – some justification of this choice, or a demonstration that using exclusive/non-overlapping boundaries for each bin would help clarify, if this is a correct interpretation.

We indeed used overlapping wedges by using 45-degree feature wedges and going in steps of 10 degree. This ensured sufficient data in each reference (“training”) bin, while yielding smooth tuning profiles. (This is conceptually very similar to the use of overlapping time windows in a time-frequency analysis where the same data is used in multiple overlapping time windows.) Importantly, however, as shown below in Reply Figure 1, nearly indistinguishable results were obtained when no overlap was allowed (in the case below, we used 30-degree reference wedges to allow a better trade-off between tuning resolution and trials per wedge).

Reply Figure 1. Tuning profile (all targets) and summary decoding statistics (per condition) when no overlap is allowed between neighbouring orientation bins. Horizontal lines again show significant clusters of the main cueing effect (blue), main distractor effect (magenta), and their interaction (green).

We have also clarified our Methods sections in accordance:

p19 (Methods): “For each bin (i.e., each orientation wedge), we included reference trials within ± 22.5 degrees of the bin’s center. While we thus allowed substantial overlap between our bins (yielding smoother tuning profiles), we confirmed that highly similar results were obtained when not allowing any overlap between reference-bins.”

8. How will the data/code be shared?

We intend to share the experiment presentation script, the behavioural and EEG data, and the essential analysis code upon publication. We have now also added a statement regarding this to our Methods.

Minor

1. In the abstract, there's a small typo: "but, instead, delay[s] it"

Thank you, we have corrected this.

2. Fig. 1c – is it possible to break out the overlaid 'fidelity' plot (which is on top of the pattern similarity image) into a separate panel? This panel is especially dense at present, and readers may benefit from smaller chunks of digestible data. Additionally, adding a color bar for panel 1c would also help readers interpret the data.

Thank you, we have now broken this down into separate panels, alongside other revisions to this figure (now Figure 2) as prompted by further comments. This Figure now looks like this:

Figure 2. Time-resolved EEG orienting decoding of targets and distractors. (a) Time resolved orientation tuning profiles. Data represent the mean-centred pattern similarity (quantified using the *Mahalanobis* distance) between the test trials and the reference trials, as a function of the angular difference between test and reference trials (y axis). The inset in the leftmost panel highlights the 8 electrodes that were used for the orientation decoding analysis. **(b)** Average tuning profiles for the data in panel **a**, in five successive time windows. **(c)** Timecourses of the corresponding summary decoding statistic (Methods for details). **(d)** Time resolved decoding (summary statistic) as a function the EEG electrode row used for decoding. Topography plot to the right shows the same data in a more conventional manner whereby the value in each electrode indicates how well the row to which that electrode belongs is able to decode target orientation. Error bars represent ± 1 s.e.m calculated across participants ($n = 30$).

3. Fig. 1c – can this analysis be shown separately for target representations (on no-distractor trials) and distractor representations (on distractor+ trials)?

As shown above, we have now separated this out by target decoding in distractor-absent trials, target decoding in distractor-present trials, and distractor decoding. We hope that this will help emphasize the utility of our decoding approach for individuating targets and distractors (even when these are presented in close temporal proximity) and tracking each in time. Only after presenting and discussing this important feature of the approach, do we turn to the influence of cues (which is our main interest).

To accommodate these revisions, we have also substantially reworked the associated text in the Results section.

4. Fig. 4a-b – again, very dense figure panels. If possible, separating the topographies and scatterplot from the TFR would make these figures easier to digest, and features of the data easier to see. This would also avoid the problem of covering up portions of panel a with the topographies.

We agree, and have now placed the topographies and scatterplots below the TFR plots (see current Figure 6, also to be found below a related comment from reviewer 2).

5. Are trials culled based on eye movements? The methods only describe removing trials with identified blink artifacts

Thank you, we have now added the following explanation to our Methods:

p18 (Methods): “We did not explicitly cull for eye movements as the task was presented at fixation (although trials with large artifacts as a result of saccading would have likely been removed anyways based on our variance-based artifact rejection).”

Reviewer 2

Using EEG decoding methods in a visual orientation reproduction task with preparatory auditory cues and visual distractors, the manuscript shows that the representations of target and distractor orientations in broad-band time-domain EEG signals (ERPs) are affected by the preparatory cue in two specific ways: (1) enhancing the representation of the target stimulus; and (2) delaying the interference of distractors with target representations. These two changes in EEG target decoding by the auditory cue are then related to the attenuation of posterior alpha oscillations. The authors conclude that anticipation is mediated by alpha oscillations through target representation enhancement and delayed distractor interference.

In my view, the data provided in this manuscript is insufficient to advance this conclusion. I have three main objections:

Thank you for your valuable comments. As you will see below, we have made several important clarifications and have added the outcomes of several additional analysis to our manuscript. Before turning to more elaborate responses to each of the points below, we first wish to emphasize that our main question regards how anticipatory cues facilitate target processing in the face of temporal distractors. The analyses linking the identified modulations (enhancing target decoding and delaying distractor interference) also to states of attenuated alpha oscillations are “only” subsidiary, and we now more carefully outline their role in our manuscript as well as the conclusions we associate with them (as also described in response to point 3 below). We further wish to note that our focus on how preparatory cues can overcome temporal distractors provides the major novelty and advance in relation to previous work (e.g., Kok et al., 2012), in addition to the fact that we capitalised on the high temporal resolution of EEG to evaluate at what processing stages cues impact target representations (as further elaborated on in response to point 2 below).

1) The preparatory cue is shown to improve behavioral performance in this task, but none of the brain signal analyses are then shown to be directly related to this behavioral effect. The EEG differences observed between cued and uncued trials could be related to multisensory processing, divided attention, and not specifically to anticipating visual perception. This is particularly disappointing given the fact that the decoding methods used can power single-trial analyses (see Fig. 4d) and thus relate to behavioral errors even on a trial-by-trial basis (see for instance Kok et al. 2012). Such approaches could be used at several points in the manuscript to validate the interpretations on the relation of EEG signals with behavior. For instance, the 2 mechanisms identified in Fig. 3 could be specifically linked to reproduction error by restricting the analysis to uncued trials and checking correlation between cue decoding and reproduction error, and distractor interference latency and reproduction error. Similarly, the relationship of alpha power with decoding identified in Fig. 4 should also correlate with reproduction error to validate the current interpretation of the manuscript.

As shown in added Figure S6 below, the correlations between trial-by-trial variability in target decoding and trial-by-trial variability in behavioral performance were not reliable. As we elaborate below, many features in our task may have contributed to this. Therefore, while we agree that the demonstration of such trial-wise correlations would have provided additional support for our conclusions, we disagree that the validity of our conclusions are contingent on the presence of such correlations. There are many potential reasons why we might not have been able to observe such correlations in our data. Foremost, in our task, behavioural performance on single trials is likely influenced by many additional factors that are not captured by the early EEG responses that we focused on. Examples are post-target lapses in short-term memory (response probes appeared only 500 ms after the target), post-target changes of mind as to which of the two items was the target, perceptual interference by the probe stimulus, motoric errors, and so on. While the variability caused by such factors is likely to have “averaged out” in the comparison between condition-averages, it may have substantially hampered our ability to observe robust trial-wise correlations between decoding and behavioral performance measures (note that these later, post-encoding, processes are irrelevant to

the correlation between pre-target alpha amplitude and decoding, which may explain why this correlation did work). We should also note that our experiment was not geared toward looking at trial-wise correlations but, rather, was set up to compare experimental conditions with each other. Possibly, by incorporating multiple conditions, our correlational analyses (within each condition) were underpowered. Furthermore, we cannot rule out the presence of one or more “third variables” that may influence both decoding and performance, but in opposite ways – thereby possibly countering their estimated correlation.

It must however be emphasized that the pattern of decoding and behavioral performance did correspond really well when comparing condition-averages – both measures showed better performance for cued trials, larger cueing effects for distractor-present trials, prominent interference by the presentation of distractors, and the largest interference for the earliest distractors (for which we have now included the decoding results in Fig. S3, also to be found below a related comment below).

We do contend with both reviewers that this is nevertheless important information to share. To be more transparent about this aspect of our data in our manuscript, we have therefore now added the following:

p11 (Results): “**Correspondence between decoding and behavioural performance**

In a simple perceptual task one would expect the quality of target decoding to correlate with behavioural performance on a trial-by-trial basis. In our task, however, many factors are likely to influence behavioural performance, of which the perceptual processing that takes place during initial encoding is but one. Indeed, behavioural performance on single trials is likely influenced by many additional factors that are not well captured by the early EEG responses that we focused on (such as post-target lapses in short-term memory, post-target changes of mind as to which of the items was the target, perceptual interference by the probe stimulus, motoric errors, and so on.). This may explain why we were not able to demonstrate compelling and consistent correlations between the trial-by-trial variability in the magnitude of target decoding and in behavioural performance (Fig. S6). Furthermore, we note that the experiment was designed to compare decoding and performance among conditions, and not for maximising variability within the conditions. This will have further compromised the sensitivity for such correlations. However, it should at the same time be noted that, at the level of condition averages, the patterns in target decoding and in the behavioural reproduction accuracy showed excellent correspondence – both measures showed better performance for cued trials, larger cueing effects for distractor present trials, prominent interference by the presentation of distractors, and the largest interference for the earliest distractors.

Figure S6. Trial wise correlations between target decoding and behavioural performance. Time-resolved trial-wise correlations between target decoding and reproduction error (left) as well as reaction time (right), as a function of cue and distractor presence. Note that we hypothesised negative correlations, provided that better behavioural performance is associated with lower reproductions errors and lower RTs. The lines show the group-mean correlation values and the shadings represent ± 1 s.e.m. calculated across participants ($n = 30$).

In addition, we could confirm that the size of the cue-induced alpha attenuation correlated with the size of the relative RT facilitation by the cue across participants. We have added these results to updated Figure 6 (panel b, also depicted below), and have added associated text to our Results section:

p9 (Results): “Further corroborating this attentional interpretation, we also found that those participants who showed the largest cue-induced alpha attenuation, also showed the largest cue-induced reductions in RT (expressed as the RT-ratio between cued and uncued trials), thus yielding a positive correlation (Fig. 6c; cluster $p = 0.005$, cluster interval in the non-permuted data: -180 to 220 ms post target; frequency range: 6 to 11 Hz). This correlation has a clear posterior topography (topography Fig. 6b).”

Figure 6. Attenuated posterior alpha oscillations predict enhanced target decoding (across participants) and distractor resistance (across trials). (a) Time-frequency plot of the cue-induced modulation in spectral power, expressed as a percentage change (i.e. $[(\text{cued} - \text{uncued}) / \text{uncued}] * 100$). Data from all posterior electrodes marked in the inset in the right top. Topographies show modulations from 5 to 10 Hz in the interval between -400 to -200 ms (left) and from 8 to 14 Hz in the interval between 0 and 300 ms post-target (right). Topographies were scaled according to the same colorbar as the time-frequency plot. (b) Time-frequency plot of the correlation (across participants) of the cue-induced modulation with the magnitude of the main cueing effect on reaction time (RT; expressed as a ratio between cued and uncued RTs). (c) Similar to panel b, except showing the correlation with the main cueing effect on target decoding (averaged over 118 to 248 ms post-target; see Fig. 4a) The participant-specific magnitudes of the alpha modulation used for the scatter plots in panels b and c were extracted from the significant time-frequency clusters and only serve to show the underlying distributions. (d) Time courses of target decoding in uncued trials as a function of distractor presence and pre-target alpha amplitude (median split). Trials were sorted by alpha amplitude averaged over all posterior channels in the 500 ms pre-target interval. Inset shows associated pre-target spectra. (e) Distractor interference time courses as a function of pre-target alpha state. Same conventions as for Figure 4b. (f) Time courses of the trial wise correlation between pre-target alpha amplitude and target decoding, separately for distractor-present and absent trials. Shadings represent ± 1 s.e.m. calculated across participants ($n = 30$). The green shaded band in panel f highlights the similarity of the alpha-dependent decoding effect with the cue-dependent decoding effect (the interaction effect) in Figure 4.

Finally, regarding the potential cognitive factors that might mediate the cueing effects on target decoding (and the identified interaction with distractor presence), we should note that the influence of the cues appeared largely specific to the targets and was particularly pronounced in distractor-present trials. This shows that the cues did more than merely boost all sensory information in a non-selective way (i.e., equally for both target and distractor). Rather, they specifically helped separating targets

from distractors in time (an ‘attentional’ function). We have added this point to the relevant paragraph of our Discussion (page 13). While it remains true that there are many potential ways to label the cognitive processes that mediate this (which we acknowledge at several instances in our manuscript), in order to explain our data these must run through anticipatory states (and be to some degree correlated with alpha-oscillatory markers of preparatory attention); and this is all we claim.

The behavioral impact of the visual mask is also known to be different depending on the similarity of target and distractor (Magnussen et al. 1991; Magnussen and Greenlee 1992, 1999; Rademaker et al. 2015), this could also be validated in their behavioral data and then used to test if distractor orientation difference affects target decoding in a direction consistent with the behavioral effects.

Firstly, thank you for bringing to our attention this relevant line of research. As also prompted by reviewer 1, we feel this work on distraction in visual working memory is relevant to our study and have now added a paragraph dealing with this to our discussion (also pasted below the relevant comments from reviewer 1).

In addition, we have now also quantified behavioral performance as a function of target-distractor similarity. As shown in Reply Figure 2 below, responses appeared “pulled” toward distractor orientations (and more strongly so in uncued trials). At the same time, however, mixture-modelling of the behavioural data (as advised by reviewer 1) also revealed a large proportion of ‘swapping errors’ in which participants reported the distractor orientation instead of the target orientation (and further showed that this too was significantly reduced by the cue, as now brought forward and quantified in the revised manuscript). We have incorporated these mixture-modelling results in our manuscript in updated Figure 1 alongside an added paragraph to our Results section (both to be found below the relevant comment of reviewer 1). This large proportion of swapping errors complicates further analysis of this apparent “pull” effect, as the latter effect is likely a reflection of these swaps. Following these new insights, we also evaluated whether target and distractor decoding differed between swap and non-swap trials but again found no compelling differences (likely due to the same reasons as listed in our response to the first comment regarding correlations with performance, in addition to the difficulty of reliably classifying individual trials as swaps).

Reply Figure 2. Response deviations occur in the direction of the distractor orientation. However, this is likely the result of the larger proportion of ‘swap errors’ (that were also reduced in cued compared to uncued trials); see updated Fig. 1. Data from distractor-present trials with the 100-ms ISI. Dashed lines represent ± 1 s.e.m.

Also the ISI dependence of the behavioral effect in Fig. 1b could be used to test the relationship of target decoding effects and behavior.

Thank you for this interesting suggestion. We had concentrated our efforts on the 100-ms ISI condition, as this contained the vast majority (80%) of distractor-present trials. We initially only included the 20 and 200-ms ISI conditions to map out the basic behavioral effect of ISI (that we had also observed in prior pilot runs of the task). For the more sophisticated decoding analyses, we trial numbers in these conditions were deemed to be insufficient to support robust statistical analyses. Still, because we agree that the descriptive patterns in these these data may be of interest too, we have now also quantified target decoding in these trials and have included this in Supplementary Figure 3 (also pasted below). Importantly, although we could not observe significant (cluster corrected) cueing effects in these trials (likely due to low statistical sensitivity as a consequence of low trial numbers), we did observe qualitatively similar effects of the cue on protecting the target from distractor interference (i.e., delayed distractor interference). As also emphasized in response to comment 1, these data also again confirmed a nice correspondence between target decoding and behavioural accuracy at the level of condition-averages. For example, they suggest that distractor interference much larger in the 20-ms ISI condition, and that cueing benefits are particularly prominent for the 20 and 100-ms ISI conditions, but only moderate or absent for the 200-ms ISI condition (as was the case for the cueing benefits on behavioural accuracy in face of these different distractor times). We hope that this, together with our other new complementary analysis (discussed before our point-by-point replies), further increases trust in our main decoding results.

We have added the following to our Results and Supplemental Information:

p8 (Results): “Although limited trial numbers for the 20-ms and 200-ms ISI distractor conditions hampered statistical sensitivity for quantifying cueing benefits on decoding, we did observe qualitatively similar patterns in these conditions whereby, descriptively, distractor interference immediately after the respective distractor time appeared attenuated by the cue (Fig. S3; note that we again only included distractor-absent trials in the reference sets, to ensure the same reference for all three ISI conditions). This appeared particularly clear in the 20-ms ISI condition for which we also observed a similar cueing benefit on behavioural accuracy. In this condition we further noted substantially larger distractor-interferences in target decoding (Fig. S3) in further agreement with the behavioural performance data.”

Figure S3. Target decoding in cued and uncued distractor-present trials as a function of the interval between target and distractor. As in Figure S2, we only included distractor-absent trials in our reference sets to ensure a fair comparison between the 20, 100, and 200-ms ISI conditions. For the middle (distractor interference) panels, we subtracted the decoding in the distractor-absent trials (unshown) to yield the same type of distractor interference plot as in Figure 4b. Arrows indicate effects of interest, whereby distractor interference appears delayed in cued compared to uncued trials. The right panels show the difference between cued and uncued trials. Black vertical lines indicate the onset of the distractor in the different conditions. Shadings represent ± 1 s.e.m. calculated across participants ($n = 30$).

An additional component of their task that could provide behavioral parameters to relate with the subjects' anticipatory state is the ITI. Despite the flat hazard rate defined for the ITI, it is likely that there is a subjective urgency that can have behavioral impact (Janssen and Shadlen, 2005) and impact correspondingly ERP decoder modulations.

Thank you for this suggestion. We aimed to minimize effects of hazard-rate by always drawing ITIs from a (truncated) negative exponential distribution that approximated a flat hazard rate (as also stated on page 17 in our Methods section). This is a key difference with the study by Janssen and Shadlen (2005) (as well as Schoffelen et al., 2005; and others) where the authors explicitly manipulated distinct hazard rates that yielded clear expectation profiles and that could be compared against each other.

Prompted by this suggestion, we nevertheless investigated the potential influence of ITI but could not find a strong influence on performance (see Reply Figure 3 below). In a way, the results below thus show that our aims to minimize the influence of hazard-rate were successful.

Reply Figure 3. Behavioural performance as a function of the preceding inter-trial-interval (ITI). Data were binned in 10 successive bins of ITI, and centred on the average ITI in that bin. Dashed lines represent ± 1 s.e.m.

2) The association of alpha-band activity with the two EEG mechanisms (ERP target decoding accuracy and distractor interference latency) is sketchy and not consistent. Figure 4c-d show nicely how pre-stim alpha power acts very similarly to the anticipatory cue, in relation to the distractor interference effect of Fig. 3. However, these same panels show that pre-stim alpha does not reproduce the main effect of the auditory cue (main effect of cue in Fig. 3c). This is in contrast with the message of Fig. 4b. Are there 2 alphas? A cue-induced alpha that is responsible for the main effect of cue, and a spontaneous alpha that accounts for the interaction effect (distractor interference)? While the spontaneous alpha analysis in Fig. 4c-d is very suggestive, the interaction effect reported in Fig. 3 is triggered by the cue presentation, so it should also be present in the analysis of cue-induced alpha. Does the cue-induced alpha correlate in an inter-individual analysis with the magnitude of the interaction effect? In Discussion it is suggested that both spontaneous and cue-induced alpha correlate

with the main effect and the interaction effect ("This was the case both for the task-related modulation..."), but this is currently not supported by Fig. 4.

Thank you, you are right. We now make this distinction between the across-participant and across-trial results much more explicit. For example:

p12 (Discussion): "[...] This was the case both for the task-related modulation by anticipatory cues (across participants), as well as for the spontaneous fluctuations in alpha amplitude in the absence of cues (across trials) – although we note that the variability in the cue-induced modulation across participants only correlated significantly with the main cueing effect on decoding, whereas the spontaneous variability across trials only correlated with the target decoding in distractor-present trials, in the identified interaction window. Resolving this apparent discrepancy remains an interesting target for future research as it suggests there may be distinct sources of variability in posterior alpha oscillations that may have different bearings on perception."

We must also point out that our main contribution relates to the observed differences between experimental conditions presented in Figure 3 (now Figure 4). This alpha-based analysis, while clearly of interest, involves a subsidiary analysis aimed merely to find additional support for the observed differences and to link this work to the attentional literature where alpha modulations are commonly reported. While we agree (and now make more explicit in our manuscript) that the alpha-based correlations did not correspond one-to-one with the observed cueing effects on decoding (with the across-participant correlation only corroborating the main cueing effect, and the across-trial correlation only corroborating on the interaction effect), they do provide at least some tentative evidence that the observed cueing effects are linked to (and may at least partially be mediated by) states of attenuated alpha oscillations.

An additional aspect of alpha-band activity that is not addressed in this analysis is the role of alpha phase. Recent studies are showing that alpha paces the sequence of perceptual cycles (VanRullen 2016) and that auditory stimuli reset the posterior alpha rhythm, with perceptual consequences (Romei et al., 2012). The timings and intervals of interest in the current task are within one alpha cycle, possibly making the effects sensitive to alpha phase. Could a sizable part of the behavior and target decoding results be explained by the alpha phase-resetting of the auditory cue?

This is an interesting point. Unfortunately, potential phase-resetting of posterior alpha oscillations by the auditory cue was hard to detect in our task. As shown in Reply Figure 4 below, inter-trial coherence (a measure of phase-alignment across trials) was dominated by the auditory evoked response (possibly overshadowing more subtle effects on posterior alpha oscillations) and later by the ERP to the visual target itself. Moreover, our cue-target interval was only 500, thereby possibly not allowing sufficient time to detect the potential phase-resetting of the ongoing alpha oscillations in the interval between the cue the anticipated target.

Reply Figure 4. Inter-trial coherence in cued and uncued trials robustly tracks the auditory cue and visual target, but shows no clear signs of anticipatory phase-resetting of posterior alpha oscillations. Data are averaged over the posterior channel cluster that was also used in the alpha-amplitude based analysis in updated Figure 6.

We nevertheless explored target decoding as a function of pre-target alpha phase (e.g., by sorting trials by pre-target alpha phase, and evaluating circular-to-linear correlations), but were not able to find compelling phase-dependence of decoding either. Still, because we consider this an important avenue for future research, we have added the following to our manuscript:

p11 (Results): “We also investigated potential cue-induced resetting of oscillatory phase (as in e.g., Romei et al., 2012), as well as potential relations between pre-target alpha phase and target decoding, as the phase of alpha oscillations may also critically shape perception (VanRullen, 2016 for review). However, beyond a clear phase-reset in the lower frequencies (which, again, most likely reflected the auditory ERP), we did not find compelling evidence for anticipatory phase-alignment following the cue, nor did we observe compelling associations between pre-target phase and target decoding. However, as our task might not be ideal for looking at this, this should remain an interesting avenue for future endeavours.”

3) out of the 2 mechanisms identified in this manuscript, the main effect of the cue has already been reported before (Kok et al 2012). The manuscript does not discuss this influential, recent result at all.

Thank you. We should have related our results explicitly to the results in Kok et al (2012). Because we were specifically interested in the effects of cued preparation on overcoming temporal distraction and were using high temporal-resolution methods accordingly, Kok’s relevant study was not on the forefront of our thinking, but the reviewer is correct that it still has relevance. We must, however, emphasize that our observations make two substantial advances over the previously published results by Kok et al. First, we manipulated the presence of temporal distractors and investigated not only how cues affect target decoding, but also how they enabled to separation of targets from distractors that compete within the time frames of attentional competition. Second, we decoded target identities from high temporal resolution EEG (instead of fMRI), which enabled us to temporally define the amplification of target decoding and show that this occurs already in early visual responses within the first few hundred millisecond of processing. We further note that, in Kok et al., expectations regarded the orientation of the upcoming target, whereas in our study cues only informed target timing (i.e., that a target would occur in 500 ms), without giving any information as to the identity of this target.

We now cite this relevant prior work and highlight what additional insights are gained from our manuscript:

p11 (Discussion): “Previous fMRI work has already demonstrated that anticipatory expectations about features defining target identity can increase representational information in human visual cortex (e.g., Kok et al., 2012). Our results show that even simple anticipation of stimulus timing, with no expectation that enables any feature-related template to be established, also significantly boosts target representations. Furthermore, by resorting to high temporal-resolution EEG measurements, our results reveal that this occurs already during early sensory processing stages. Specifically, this “representational boost” peaked around the classical N1 time range.”

p12 (Discussion): In addition to a direct influence of anticipatory cues on target processing, we also observed a second effect that depended on distractor presence (thus further complementing previous fMRI work on the influence of expectations on target representations as in Kok et al., 2012).

In addition I have a number of methodological concerns:

1) “Participant-specific trial-averaged ERP and decoding time courses were subsequently smoothed with a Gaussian kernel with a 15 ms standard deviation”. I miss a strong argument to perform such operation in the final average. This looks like a “cosmetic” preprocessing step and “it obscures the

ability to evaluate the physiological plausibility of an effect, and thereby hides relevant complementary information from readers” (Van Ede & Maris 2016).

Thank you for raising this point. When Eric Maris and I (first author of the current article) put together this opinion piece on data reporting practices, we were particularly concerned with practices in which data are only reported in “collapsed form”, such as in a single bar graph, when the data underlying that bar graph contains multiple dimensions (e.g., time, space, and frequency). In our view, we adhere to these ‘guidelines’ by reporting the decoding as a function of both time and space, and by showing time, frequency, and spatially-resolved correlations between oscillatory amplitude and decoding.

Our data smoothing step was in no means intended to cover up or obscure any aspect of our data. Also, I believe in our opinion piece, we did not argue against data smoothing as a bad practice. In fact, data smoothing can substantially increase statistical sensitivity by reducing variance across participants. For example, if participants show a qualitatively similar effect of the cueing manipulation on the target decoding, but in slightly different latencies (e.g., because their ERP components have slightly difference latencies due to slight anatomical and physiological differences) then temporal smoothing can allow to “bridge” such differences and thereby increase second-level group statistics. Because of these important advantages, data smoothing is a very common practice in both fMRI (spatial smoothing) and EEG/ERP research (temporal smoothing; as also comes with low-pass filtering of ERP components).

We quantified stimulus identity decoding in a timepoint-by-timepoint manner, whilst preserving the 1000 Hz sampling rate of the data. We then applied a smoothing kernel to the resulting decoding time courses simply with the aim of reducing inter-timepoint and inter-subject variability, and thereby to increase statistical sensitivity.

We now added the following to clarify this point:

p18 (Methods): “This allowed to bridge variability in the timing of the responses across participants, without smoothing away the essential characteristic of the ERP waveform (i.e., the distinct peaks). Such smoothing has similar consequences as low-pass filtering, which is also common in ERP research.”

Of course, it is still true that the decision for the amount of smoothing is relatively arbitrary, and that data can also easily be “over-smoothed”. In this light, it is important to note that with the applied amount of smoothing, our ERPs still showed clearly discernible components. To further take away the concern that we may have “cherry-picked” our particular smoothing parameter and to further increase transparency, we now show our main contrasts of interest as a function of data smoothing (Fig. S1, also pasted below) and show that our main decoding as well as our ERP effects are largely invariant to the choice of smoothing (within reasonable ranges, here evaluated for 6 steps between 5 and 30 ms). We also explicitly point to this at the relevant section in our manuscript:

p6 (Results): “We also note that each of these three effects were largely invariant to our choice of data smoothing (Fig. S1).”

p18 (Methods): “We did confirm that our main results were largely invariant to this particular choice of smoothing (Fig. S1).”

Figure S1. Decoding and ERP effects are largely invariant to choice of smoothing kernel. Main effects of cue presence (left), distractor presence (middle), and their interaction (right) for both target decoding (upper) and ERPs (lower) as a function of the width (in standard deviations) of the applied Gaussian smoothing kernel (different colors). Plotted are the t-values associated with the simple contrasts (left: cued vs. uncued; middle: distractor present vs. absent; right: cued minus uncued in distractor-present vs. distractor-absent trials; see also Figure 4c). All results reported in the manuscript were based on the smoothing kernel with the 15 ms standard deviation (i.e., the red line).

2) "Significant clusters". Wrong concept that appears repeatedly in the manuscript (Figure 3 and Figure S1 captions; “three significant clusters”, page 4, etc). The non-parametric cluster-based permutation test serves to test a null hypothesis: The data (not the parameters estimated from the data) in different experimental conditions came from the same probability distribution, so they are exchangeable. The alternative hypothesis consists that the data in different experimental condition do NOT come from the same probability distribution. Stating “there is a significant cluster” is simply wrong. The statistical significance indicates an informed decision about the uncertainty to accept or reject the null hypothesis but never about “when” (time) or “where” (topography, frequency) those differences take place. The correct statistical conclusion would be that the authors have found a significant difference between condition A v.s. condition B. I encourage the authors to revisit Maris & Oostenveld (2007) and Maris (2012). Check: http://www.fieldtriptoolbox.org/faq/how_not_to_interpret_results_from_a_cluster-based_permutation_test

Thank you for reminding us of this point regarding statistical reporting practices. We agree that we were sloppy in our initial reporting of this. We have now revised our manuscript in the following ways. First, we make clear that our evaluation involved three separate comparisons, and explicitly state what inferences are and are not warranted with this statistical approach:

p6 (Results): “Cluster-based permutation statistics (Maris and Oostenveld, 2007) were used to evaluate the main effects of cueing (cued vs. uncued trials, both for target and distractor decoding), distractor presence (distractor-present vs. absent trials), and their interaction (i.e., the cueing effect in distractor-present vs. absent trials), while circumventing the multiple-comparisons encountered along the time axes. Although we below state the time-ranges of the significant clusters as they were observed in the observed (non-permuted) data, it is relevant to note that this cluster-based permutation test does not warrant inferences regarding exact time ranges that are significant, as it only evaluates whether the compared conditions are ‘exchangeable’ or not – and, for this evaluation, it considers the full time range (Maris and Oostenveld, 2007 and Maris, 2012).”

Second, while we still state the time windows of the largest clusters as they were observed in the non-permuted data, we explicitly state that these involve the ranges in the non-permuted data (e.g., “cluster interval in non-permuted data: 118 to 248 ms post target”). We do so at every relevant instance, including in the relevant figure captions (“Horizontal lines mark where the clusters of the contrasts that survived cluster-based permutation statistics were observed in the non-permuted data”). Together with the added text above, we hope this makes sufficiently clear that our statistical inferences do not concern these time ranges, but that these ranges were simply where the largest clusters were located in the observed data.

3) EEG orientation decoding (in Methods). The authors state that “a 250 ms pre-target baseline was subtracted”. Was it performed at a single-trial level? Such a short period will contain a lot of noise and a demeaning operation (mean subtraction of the entire epoch) would be way more efficient. If data is stationary, single-trial baseline correction would correspond to baseline correction averaging the data across trials. Please check the stationarity of your data: do you find significant differences in your ERP and decoding plot subtracting the single trial baseline estimates v.s. the grand-mean baseline subtraction? If the answer is positive, please consider a more careful data normalization (see Grandchamp & Delorme 2011).

Apologies for not stating this clearly. We now make clear that we subtracted trial-specific pre-target baseline values in the time-domain (which is standard practice in ERP research, at least in our lab). As far as we can tell, it appears the reference by Grandchamp and Delorme (2011) regards the use of baselining when quantifying spectral perturbations, where it is also common to use relative changes from baseline. We only used baselining for our time-domain signals (that we subjected to our decoding and ERP analyses) where absolute changes are the standard. For the only time-frequency-analysis that we presented (regarding the alpha modulation; updated Figure 6), we directly contrasted cued and uncued conditions without baselining at all.

While subtracting the mean of the entire epoch would also have been a possibility, we feel that the downside of this for characterising (single trial) ERPs (that could then be subjected to our decoding pipeline) is that this can induce carry-over effects between components (e.g., if a given trial has a large positive deflection at time X, a demeaning operation will introduce relative down weighting of values at time points other than X). This is avoided by subtracting a pre-stimulus baseline.

Minor comments:

Please provide the number of trials per condition left after artifact rejection. This is important information to interpret the decoding results and to design future experiments based on your findings. As much as possible, authors should follow well-established guidelines (Keil et al., 2014).

Thank you. We have now added this information to our Methods:

p18 (Methods): “After artifact rejection, there were 1042 ± 24 (mean \pm 1 s.d.) trials left. Broken down by our main four conditions, numbers were: 291 ± 8 (cued, no distractor), 288 ± 7 (uncued, no distractor), 233 ± 5 (cued, distractor at 100-ms ISI), and 230 ± 7 (uncued, distractor at 100-ms ISI).”

Figure 1c: There is no color bar associated. Please add the proper color bar to the figure or specify in the figure legend that Figure 1c and d share the same color bar.

Figure 1c summary decoding statistic is not clear. The y-scale is difficult to read and this statistic is the one employed throughout the paper. I think it deserves a subfigure in its own right. Figure 2bc in Wolf et al., (2017) could be a good choice.

Thank you. We have now revised this Figure (Figure 2 in the revised manuscript) to incorporate these valuable suggestions (and have also made some further changes following comments from reviewer 1). This revised Figure (also pasted below) should also help better emphasize the utility of our decoding approach for individuating targets and distractors and tracking each in time.

Figure 2. Time-resolved EEG orienting decoding of targets and distractors. (a) Time resolved orientation tuning profiles. Data represent the mean-centred pattern similarity (quantified using the *Mahalanobis* distance) between the test trials and the reference trials, as a function of the angular difference between test and reference trials (y axis). The inset in the leftmost panel highlights the 8 electrodes that were used for the orientation decoding analysis. (b) Average tuning profiles for the data in panel a, in five successive time windows. (c) Timecourses of the corresponding summary decoding statistic (Methods for details). (d) Time resolved decoding (summary statistic) as a function the EEG electrode row used for decoding. Topography plot to the right shows the same data in a more conventional manner whereby the value in each electrode indicates how well the row to which that electrode belongs is able to decode target orientation. Error bars represent ± 1 s.e.m calculated across participants ($n = 30$).

Figure 3a: the legend is confusing and it is not well explained. For example, the black line “distr –“ never appears in the figure. Same goes to Figure S1a. Please clarify this because takes time to understand your key figures.

Thank you. We agree our labelling was potentially confusing, so we have now changed the relevant legends and directly link each condition to a separate line in the legend. For this we use “c+” for cue present, “d-“ for distractor absent, and so on. We also make this clearer in the relevant Figure captions. For example in the caption of Figure 4a (previous Fig. 3a) we state: “Time courses of target and distractor orientation decoding (summary statistic) as a function of cue presence (blue for cued, “c+”; red for uncued, “c-“) and distractor presence (solid for distractor-absent, “d-“; dashed for distractor-present, “d+”).”

Figure 4b,c inset plots are too small.

Thank you. We have now enlarged this plots and placed them below the TFR plots, as can be seen in the updated Figure 6, which is also depicted below comment 1.

Reviewers' comments:

Reviewer #1 (Remarks to the Author):

I appreciate the authors' careful attention to detail in their response to my previous comments. They have satisfied all my concerns, and I believe the resulting manuscript makes a very strong contribution to Nature Communications.

Reviewer #2 (Remarks to the Author):

The manuscript has improved significantly following the revisions introduced. There is however one concern that has not been sufficiently addressed by the authors. Specifically, my 3rd methodological concern in relation to the baselining approach was not addressed with alternative baselining methods to confirm the robustness of the results. In addition, the authors now bring up an important caveat concerning "carry-over effects between components" that in my view could be contaminating their interpretation of the data considering the fact that they took a 250ms pre-target baseline for their ERP analyses: any cue-related signals prior to the target will be artifactually carried over to later points in the trial by means of the pre-target baselining procedure. It is imperative that the results are shown to be robust to baselining the data prior to the cue, and also taking other possible baseline approaches (whole trial as in Grandchamp and Delorme 2011, or fusing baselines as in Ciuparu and Muresan, 2016 DOI: 10.1111/ejn.13179). The fact that these articles referred specifically to spectral analyses and not ERPs does not justify not controlling for the possible caveats of ERP baselining that I raised in my report.

Dear Editor,

We were pleased to see that both reviewers highly valued our previous revisions and consider our manuscript substantially improved. As you will see below, we now also address the single remaining point from reviewer 2 and confirm that our results are also largely invariant to choice of baseline.

We have meanwhile also aligned the formatting of our manuscript with the Editorial policies of Nature Communication and are ready to make our data publically available through the Dryad Digital Repository upon receiving your final decision.

We are very grateful for your time and are looking forward to your decision.

Yours faithfully,

Freek van Ede, Sammi Chekroud, Mark Stokes, Kia Nobre

Replies – round 2

Reviewer 1

I appreciate the authors' careful attention to detail in their response to my previous comments. They have satisfied all my concerns, and I believe the resulting manuscript makes a very strong contribution to Nature Communications.

Reviewer 2

The manuscript has improved significantly following the revisions introduced. There is however one concern that has not been sufficiently addressed by the authors. Specifically, my 3rd methodological concern in relation to the baselining approach was not addressed with alternative baselining methods to confirm the robustness of the results. In addition, the authors now bring up an important caveat concerning "carry-over effects between components" that in my view could be contaminating their interpretation of the data considering the fact that they took a 250ms pre-target baseline for their ERP analyses: any cue-related signals prior to the target will be artifactually carried over to later points in the trial by means of the pre-target baselining procedure. It is imperative that the results are shown to be robust to baselining the data prior to the cue, and also taking other possible baseline approaches (whole trial as in Grandchamp and Delorme 2011, or fusing baselines as in Ciuparu and Muresan, 2016 DOI: 10.1111/ejn.13179). The fact that these articles referred specifically to spectral analyses and not ERPs does not justify not controlling for the possible caveats of ERP baselining that I raised in my report.

Thank you. We too feel the manuscript greatly improved after incorporating your valuable comments and suggestions. We apologise for not having responded in sufficient detail to this remaining point during our previous revision – the large number of revisions that we made detracted our full devotion to this relevant point.

We now show that our main target decoding results are also largely invariant to choice of baseline. We show this below for six different baseline variants, and we have now also included the below figure in our Supplementary Material. In particular, we show that our main results are largely invariant to the positioning of the baseline in the pre-target or pre-cue interval, as well as to baseline duration (250 or 100 ms) and quantification (mean or median subtraction).

With respect to the concern of potential “carry-over” effects when using a pre-target (as opposed to pre-cue) baseline, we additionally clarify that cues predicted target occurrence, not target identity (our decoding variable of interest). It is therefore not possible for any stimulus orientation decoding in the post-target period to be the result of a carry-over effect from a pre-target baseline.

Added Supplementary Figure 2. Main decoding results for six different baseline variants. Baselines were either positioned pre-target or pre-cue and spanned either 100 or 250 ms. In addition to subtracting the mean EEG signal of each of each baseline, we also explored subtracting the median. All baselines were performed at the single-trial level and involved subtraction. The baseline used in all other analyses was the 250 ms pre-target baseline with mean subtraction (the second column). This baseline was chosen based on a-priori reasons – it is the closest to the target processing period of interest, whilst not containing any stimulus identity information –, and this baseline comparison was only made after all other reported analyses had already been evaluated. Shadings represent ± 1 s.e.m. calculated across participants ($n = 30$).

As shown above, we found that our analysis generally yielded highly similar patterns of target decoding for all considered baselines. We did note, however, that target decoding generally yielded slightly attenuated values when using pre-cue as compared to pre-target baselines (while baseline duration and quantification yielded near perfect replications). This is likely due to the fact that pre-target baselines are simply closer to the target processing period of interest and therefore yield more consistent signal ranges (closer baselines better mitigate potential drifts in the signal). Indeed, in a subsidiary analysis, shown in Rebuttal Figure 1 below, we confirmed that pre-target (compared with pre-cue) baselines yielded less variability in the ERP across trials within the target processing period of interest.

Rebuttal Figure 1. Pre-target compared to pre-cue baselines yield less variable EEG signals in the target processing period of interest.

When considering the main effects of cue presence, distractor presence, and their interaction (second, third and fourth row in Supplementary Fig. 2 above), we also noted highly similar patterns. Main effects of cue presence and distractor presence were virtually indistinguishable between pre-cue and pre-target baselines, and for the different baseline durations and quantifications. The interaction effect, however, became slightly less robust when moving from a pre-target to a pre-cue baseline – although it should be noted that the average size, shape, and timing of this effect were largely preserved (see overlay plot in rightmost panel of the third row) and that this effect remained highly robust for each of the pre-target baseline variants. This is, however, unlikely due to a systematic (“confounding”) reason of the baseline being positioned before or after the cue. The interaction effect involves a difference in the cueing effect between distractor-present and distractor-absent trials, and whether a trial will contain a distractor is neither known before nor after the cue (this is only known once a distractor is actually presented). Instead, the less reliable interaction effect for the pre-cue (as opposed to pre-target) baselined data is likely attributed to the fact that target decoding appears slightly less sensitive and ERP variability is higher with these baselines, as shown above.

We restricted our choice of baselines to intervals preceding target/distractor processing. Our target/distractor identity decoding analysis capitalizes on differential responses to trials with different stimulus orientations. Using a baseline period that overlaps with the target/distractor processing period may therefore subtract out such patterns of interest and may additionally “smear” stimulus decoding in time. These issues are avoided when restricting the baseline to any period before target onset (pre-target or pre-cue), as there simply is no stimulus identity information available in these data (cues predict target occurrence, not identity). Only epochs prior to target onset can thus be used safely to normalise the data, while leaving trial-specific stimulus-orientation information (our decoding variable) unaffected. Thus, while we appreciate the use of the entire epoch as another useful baseline in many instances, when it comes to decoding stimulus identity, it is our principled belief that baseline periods must always be placed before onset of the to-be decoded stimuli.

Finally, considering the baselines proposed in the articles by Grandchamp and Delorme (2011) or Ciuparu and Muresan (2016), in our reading, these clever approaches were developed for tackling potential biases that may be introduced when using relative changes (division) for data that is positively skewed. We found that, when considering ERP data, relative changes from baseline can lead to highly inappropriate scaling because the average value in the baseline (the denominator) can be very close to zero. Indeed, our time-domain data was already close to being zero-centred provided that we had applied a 0.1 Hz high-pass filter during data acquisition (which we now state explicitly in our Methods section). Absolute changes (subtraction) are thus preferred when dealing with evoked responses, as in our case. Of course, one must still be cautious of the potential contribution of outlier data points when baselining. Therefore, as a complementary way to deal with the contribution of

potential outlier data in our baselines, we also considered subtracting median as opposed to mean baseline values. As we have shown above, this yielded virtually identical results.

Following these clarifications and additions, we have also added to following text to our manuscript:

Page 10/11 (Methods): “We chose to position our baseline in the pre-target period because this interval is closest in time to the data period of interest (the target/distractor processing period), whilst not in itself containing any information regarding target/distractor identity. We did, however, confirm that highly similar results were obtained when positioning the baseline pre-cue, or when changing the duration of the baseline or subtracting the median as opposed to the mean baseline value from each trial (Supplementary Fig. 2).”

Page 4 (Results): “We also note that these effects were largely invariant to our choice of data smoothing (Supplementary Fig. 1) or baselining (Supplementary Fig. 2).”

REVIEWERS' COMMENTS:

Reviewer #2 (Remarks to the Author):

The authors' reply now addresses my last methodological concerns regarding this manuscript, which are now clearly resolved with the additional supplementary figure provided.